# Introspective Learning : A Two-Stage Approach for Inference in Neural Networks

**Mohit Prabhushankar**
Electrical and Computer Engineering
Georgia Institute of Technology
Atlanta, GA 30308
mohit.p@gatech.edu

**Ghassan AlRegib**
Electrical and Computer Engineering
Georgia Institute of Technology
Atlanta, GA 30308
alregib@gatech.edu

## Abstract

In this paper, we advocate for two stages in a neural network's decision making process. The first is the existing feed-forward inference framework where patterns in given data are sensed and associated with previously learned patterns. The second stage is a slower reflection stage where we ask the network to reflect on its feed-forward decision by considering and evaluating all available choices. Together, we term the two stages as introspective learning. We use gradients of trained neural networks as a measurement of this reflection. A simple three-layered Multi Layer Perceptron is used as the second stage that predicts based on all extracted gradient features. We perceptually visualize the *post-hoc* explanations from both stages to provide a visual grounding to introspection. For the application of recognition, we show that an introspective network is $4\%$ more robust and $42\%$ less prone to calibration errors when generalizing to noisy data. We also illustrate the value of introspective networks in downstream tasks that require generalizability and calibration including active learning, out-of-distribution detection, and uncertainty estimation. Finally, we ground the proposed machine introspection to human introspection for the application of image quality assessment.

## 1 Introduction

Introspection is the act of looking into one's own mind (1). Classical introspection has its roots in philosophy. Locke (2), the founder of empiricism, held that all human ideas come from experience. This experience is a result of both sensation and reflection. By sensation, one receives passive information using the sensory systems of sight, sound, and touch. Reflection is the objective observation of our own mental operations. Consider the toy example in Fig. 1. Given an image $x$ and an objective of recognizing $x$, we first sense some key features in the bird. These features include the color and shape of the body, feathers and beak. The features are chosen based on our existing notion of what is required to contrast between birds. We then associate these features with our existing knowledge of birds and make a coarse decision that $x$ is a spoonbill. This is the sensing stage. Reflection involves questioning the coarse decision and asking why $x$ cannot be a Flamingo, Crane, Pig or any other class. If the answers are satisfactory, then an introspective decision that $x$ is indeed a spoonbill is made. The observation of this reflection is introspection.

In this paper, we adopt this differentiation between sensing and reflection to advocate for two-stage neural network architectures for perception-based applications. Specifically, we consider classification. The sensing stage is any existing feed-forward neural network including VGG (3), ResNet (4), and DenseNet (5) architectures among others. These networks sense patterns in $x$ and make a coarse feed-forward decision, $\hat{y}$. The second stage examines this decision by reflecting on the introspective question *'Why $\hat{y}$, rather than $y_I$?'* where $y_I$ is any introspective class that the sensing network has learned. Note that there is no external intervention or new information that informs

36th Conference on Neural Information Processing Systems (NeurIPS 2022).

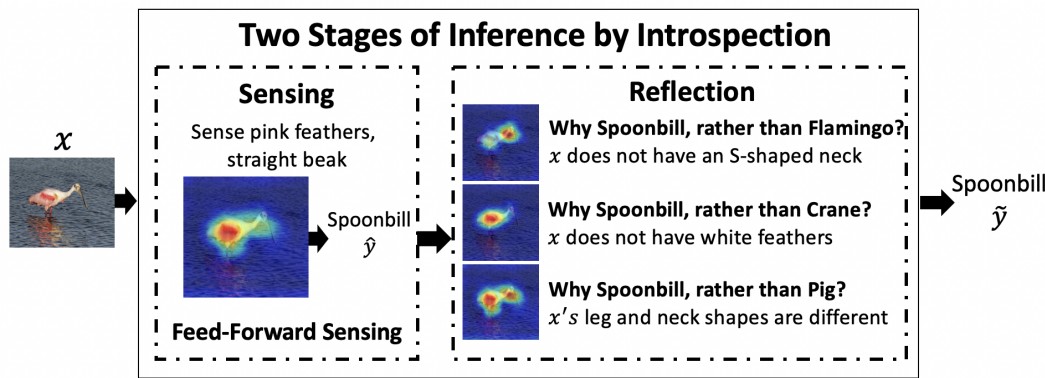

Figure 1: Example of the introspection process. The visual *post-hoc* explanations are from Grad-CAM ([6]) framework using the proposed features. Additional heatmaps along with the generation process is provided in Appendix A. The written text is for illustrative purpose only.

the reflection stage. Hence, the introspective features are *post-hoc* in the sense that they are not causal answers to the introspective questions but rather, are the network's notion of the answers. This is inline with the original definition of introspection which is that introspection is the observation of existing knowledge reflected upon by the network. These *post-hoc* introspective features are visualized using Grad-CAM ([6]) in the reflection stage of Fig. 1.

The challenge that we address in this paper is in defining the reflection stage in terms of neural networks. The authors in ([7]), utilize the same intuition to construct a *reflective stage* composed of explanations. However, when the train and test distributions are dissimilar, the predictions and hence the explanations are incorrect. We overcome this by considering alternative questions for the explanation. Consider any dataset with $N$ classes. A network trained on this dataset will have $N$ possible introspective questions and answers, similar to the ones shown in Fig. 1. Our goal is to implicitly extract features that answer all $N$ introspective questions without explicitly training on said features. This involves 1) implicitly creating introspective questions, 2) answering the posed questions to obtain introspective features, and 3) predicting the output $\tilde{y}$ from the introspective features. We show that gradients w.r.t network parameters store notions about the difference between classes and can be used as introspective features. We use an MLP termed $\mathcal{H}(\cdot)$, as our introspective network that combines all features to predict $\tilde{y}$. A limitation of the proposed method is the size of $N$ which is discussed in Sections 3.

We show that the introspective prediction $\tilde{y}$ is robust to noise. An intuition for this robustness is that not only should the network sense the feed-forward patterns, it must also satisfy $\mathcal{H}(\cdot)$'s $N$ notions of difference between classes. Hence, during inference, we extract $N$ additional features that inform the introspective prediction $\tilde{y}$. The main contributions of this paper are the following:

1. We define implicit introspective questions that allow for reflection in a neural network. This reflection is measured using loss gradients w.r.t. network parameters across all possible introspective classes in Section 3.

2. We provide a methodology to efficiently extract introspective loss gradients and combine them using a second $\mathcal{H}(\cdot)$ MLP network in Section 4.

3. We illustrate $\mathcal{H}(\cdot)$'s robust and calibrated nature in Section 6. We validate the effectiveness of this generalization in downstream applications like active learning, out-of-distribution detection, uncertainty estimation, and Image Quality Assessment in Appendix D.

## 2 Background

### 2.1 Introspective questions, features, and network

**Introspective questions**    The choice of *'Why $\hat{y}$, rather than $y_I$'?* is not arbitrary. The authors in ([8]) describe three questions that complete *post-hoc* explanations - correlation, counterfactual, and contrastive questions. These questions together allow for an alternate form of reasoning within neural networks called abductive reasoning. The sensing network predicts based on correlations.

Since, we do not intervene within the data during the reflection stage, counterfactual questions cannot be answered. Hence, we use contrastive questions as our introspective questions. Further details regarding abductive reasoning and our choice of question is provided in Appendix A.

**Gradients as Introspective Features** The gradients from a base network have been utilized in diverse applications including *post-hoc* visual explanations ([6; 9; 10]), adversarial attacks ([11]), uncertainty estimation ([12]), anomaly detection ([13; 14]), and saliency detection ([15]) among others. Fisher Vectors use gradients of generative models to characterize the change that data creates within features ([16]). ([17]) uses gradients of parameters to characterize the change in manifolds when new data is introduced to an already trained manifold. Our framework uses the intuition from ([17]) to characterize changes for a datapoint that is perceived as new, due to it being assigned an introspective class $y_I$, that is different from its predicted class $\hat{y}$. In ([18]), the authors view the network as a graph and intervene within it to obtain *holographic* features. Our introspective features are also *holographic* in the sense that they characterize the change between $\hat{y}$ and $y_I$ without changing the network. However, our features do not require interventions that become expensive with scale.

**Two-stage networks** The usage of two-stage approaches to inference in neural networks is not new. In ([7]), the authors extract Grad-CAM explanations from feed-forward networks to train a *reflective stage*. However, our framework involves reflecting on all contrastive questions rather than correlation questions. The authors in ([19]) propose SimCLR, a self-supervised framework where multiple data augmentation strategies are used to contrastively train an overhead MLP. The MLP provides features which are stored as a dictionary. This feature dictionary is used as a look-up table for new test data. In this paper, we use gradients against all classes as features and an MLP $\mathcal{H}(\cdot)$, to predict on these features. ([20]) and ([21]) consider all classes in a conditional maximum likelihood estimate on test data to retrain the model. These works differ from ours in our usage of the base sensing network. ([22]) uses gradients and activations together as features and note that the validity of gradients as features is in pretrained base networks rather than additional parameters from the two-stage networks. This adds to our argument of using two-stage networks but with loss gradients against introspective classes.

## 2.2 Feed-forward Features

For the application of recognition, a sensing neural network $f(\cdot)$ is trained on a distribution $\mathcal{X}$ to classify data into $N$ classes. The network learns notions about data samples when classifying them. These notions are stored as network weights $W$. Given a data sample $x$, $f(x)$ is a projection on the network weights. Let $y_{feat}$ be the logits projected before the final fully connected layer. In the final fully connected layer $f_L(\cdot)$, the parameters $W_L$ can be considered as $N$ filters each of dimensionality $d_{L-1} \times 1$. The output of the network $\hat{y}$ is given by,

$$y_{feat} = f_{L-1}(x), \forall y \in \Re^{N \times 1},$$
$$\hat{y} = \arg\max(W_L^T y_{feat}), \forall W_L \in \Re^{d_{L-1} \times N}, f_{L-1}(x) \in \Re^{d_{L-1} \times 1}. \tag{1}$$

Here $\hat{y}$ is the feed-forward inference and $y_{feat}$ are the feed-forward features. In this paper, we compare our introspective features against feed-forward features. Since introspection occurs after $f(\cdot)$, all our results are *plug-in* on top of existing $f(\cdot)$.

## 3 Introspective Features

In this section, we describe introspective features and implicitly extract them using the sensing network. We then analyze them for sparsity, efficiency, and robustness.

**Definition 3.1** (Introspection). *Given a network $f(\cdot)$, a datum $x$, and the network's prediction $f(x) = \hat{y}$, introspection in $f(\cdot)$ is the measurement of change induced in the network parameters when a label $y_I$ is introduced as the label for $x$. This measurement is the gradient induced by a loss function $J(y_I, \hat{y})$, w.r.t. the network parameters.*

This definition for introspection is in accordance with the sensing and reflection stages in Fig. 1. The network's prediction $\hat{y}$ is the output of the sensing stage and the change induced by an introspective label, $y_I$, is the network reflecting on its decision $\hat{y}$ as opposed to $y_I$. Note that introspection can occur when $\hat{y}$ is contrasted against any trained label $y_I, I \in [1, N]$. For instance, in Fig. 1, the network is asked to reflect on its decision of spoonbill by considering other $y_I$ that $x$ can take - flamingo, crane, or pig. These results are extracted from ImageNet ([23]) and hence, $I \in [1, 1000]$. Additional visual introspective saliency maps similar to Fig. 1 are provided in Appendix A.

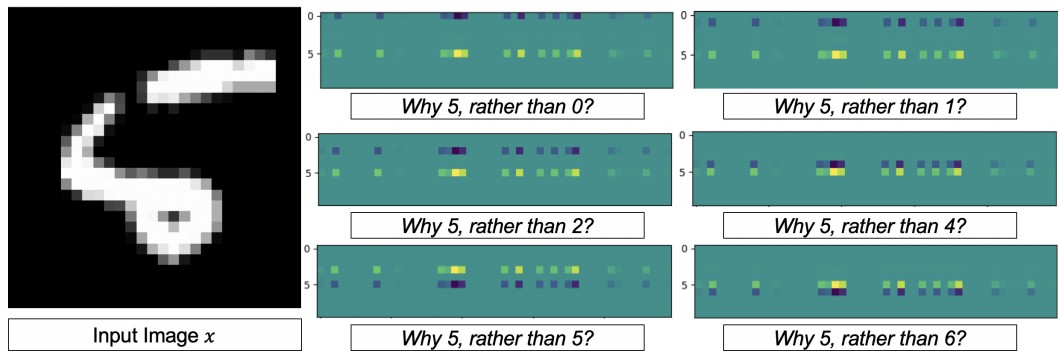

Figure 2: For the input image on the left, the $\nabla_{W_L} J(y_I, 5)$ are shown on the right. Each image is a visualization of the $50 \times 10$ gradient matrix. All images are sparse except in the prediction row 5 and introspective question row $i$.

Reflection is the empirical risk that the network has predicted $x$ as $\hat{y}$ instead of $y_I$. Given the network parameters, this risk is measured through some loss function $J(y_I, \hat{y})$. $y_I$ is a one-hot vector with a one at the $I^{th}$ location. The change that is induced in the network is given by the gradient of $J(y_I, \hat{y})$ w.r.t. the network parameters. For an $N$-class classifier, there are $N$ possible introspective classes and hence $N$ possible gradients each given by, $r_I = \nabla_W J(y_I, \hat{y}), I \in [1, N]$. Here, $r_I$ are the introspective features. Since we introspect based on classes, we measure the change in network weights in the final fully connected layer. Therefore, the introspective features are given by,

$$r_I = \nabla_{W_L} J(y_I, \hat{y}), I \in [1, N], r_I \in \Re^{d_{L-1} \times N} \tag{2}$$

where $W_L$ are the network weights for the final fully connected layer. Note that the final fully connected layer from Eq. 1 has a dimensionality of $\Re^{d_{L-1} \times N}$. For every $x$, Eq. 2 is applied $N$ times to obtain $N$ separate $r_I$. We first analyze these features.

### 3.1 Sparsity and Robustness of Introspective Features

Consider $r_I$ in Eq. 2. Each $r_I$ is a $d_{L-1} \times N$ matrix. Expressing gradients in $r_I$ separately w.r.t. the different filters in $W_L$, we have a row-wise concatenated set of gradients given by,

$$r_I = [\nabla_{W_{L,1}} J(y_I, \hat{y}); \nabla_{W_{L,2}} J(y_I, \hat{y}); \nabla_{W_{L,3}} J(y_I, \hat{y}) \ldots \nabla_{W_{L,N}} J(y_I, \hat{y})] \tag{3}$$

where each $W_{L,j} \in \Re^{d_{L-1} \times 1}$ and $r_I \in \Re^{d_{L-1} \times N^2}$. For all data $x \in \mathcal{X}$ the following lemma holds:

**Lemma 1.** *Given a unique ordered pair $(x, \hat{y})$ and a well-trained network $f(\cdot)$, the gradients for a loss function $J(y_I, \hat{y})$ w.r.t. classes are pairwise orthogonal under the second-order Taylor series approximation, each class paired with the predicted class.*

*Proof.* Provided in Appendix B.1. $\qquad\square$

**Sparsity** Lemma 1 states that backpropagating class $y_I$ does not provide any information to $W_{L,j}, j \neq I$ and hence there is no need to use $\nabla_{W_{L,j}} J(y_j, \hat{y}), j \neq i$ as features when considering $y_I$. The proof is provided in Appendix B.1. $\nabla_W J(y_I, \hat{y})$ for an introspective class reduces to,

$$\nabla_W J(y_I, \hat{y}) = -\nabla_W y_I + \nabla_W \log\left(1 + \frac{y_{\hat{y}}}{2}\right). \tag{4}$$

where $y_{\hat{y}}$ is the logit associated with the predicted class. We demonstrate the sparsity of Eq. 4 in Fig. 2. A two-layer CNN is trained on MNIST (24) dataset with a test accuracy exceeding 99%. This satisfies the condition for Lemma 1 that $f(\cdot)$ is well-trained. The final fully connected layer in this network has a size of $50 \times 10$. We provide an input image $x$ of number 5 to a trained network as shown in Fig. 2. The network correctly identifies the image as a 5. We then backpropagate the introspective class 0 using $J(5, 0)$ with $\hat{y} = 5$ and $y_I = 0$. This answers the question '*Why 5, rather*

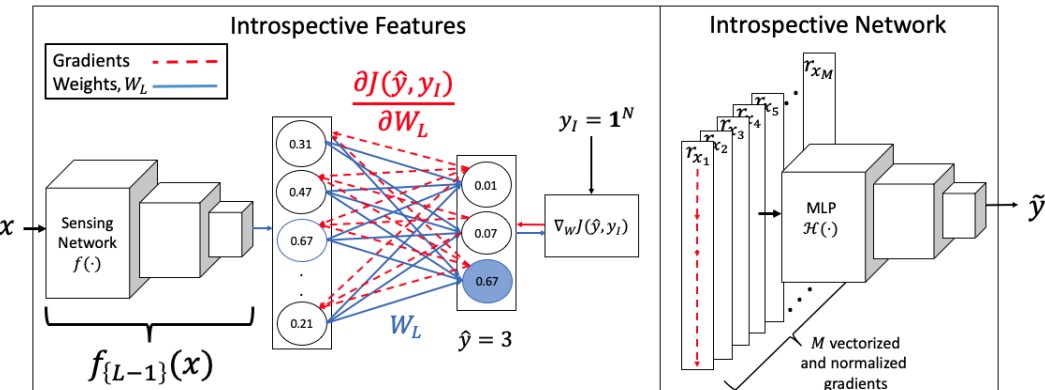

Figure 3: Introspective Learning process. Once $r_x$ for all images in the dataset are generated, an introspective network $\mathcal{H}(\cdot)$ is trained. During testing, the noisy image is passed through sensing network $f(\cdot)$, extraction module to generate $r_x$ and finally through the introspective network $\mathcal{H}(\cdot)$.

*than 0?'*. The gradient features in the final fully connected layer are the same dimensions as the final fully connected layer, $50 \times 10$ or more generally $\mathcal{O}(d_{L-1} \times N)$. This matrix is displayed as a normalized image in Fig. 2. Yellow scales to 1 and blue is $-1$ while green is 0. It can be seen that the only values present in the matrix are negative at $W_{L,0}$, in blue, and positive in $W_{L,5}$, in yellow. This validates Eq. 5 that for a fully-trained network the only values, and hence the only information, required from $W_L$ for $I = 0$ is $\nabla_{W_{L,0}}$. We show the matrix $\nabla_{W_L}$ when $I = 0, 1, 2, 4, 5, 6$. The difference among all matrices is the location of the negative values that exist at $\nabla_{W_{L,I}}$ for different values of $I$.

**Robustness** Eq. 4 motivates the generalizable nature of introspective features. Consider some noise added to $x$. To change the prediction $\hat{y}$, the noise must sufficiently decrease $y_{\hat{y}}$ from Eq. 4 and increase the closest logit value, $y_I$, to change the prediction. Hence, it needs to change one orthogonal relationship. However, by constraining our final prediction $\tilde{y}$ on $N$ such features, the noise needs to change the orthogonal relationship between $N$ pairwise logits. This motivates an introspective network $\mathcal{H}(\cdot)$ that is conditioned on all $N$ pairwise logits.

## 3.2 Efficient Extraction of Introspective Features

From Lemma 1, the introspective feature is only dependent on the predicted class $\hat{y}$ and the introspective class $y_I$ making their span orthogonal to all other gradients. Hence,

$$r_I = \nabla_{W_{L,I}} J(y_I, \hat{y}), I \in [1, N], r_I \in \Re^{d_{L-1} \times 1} \tag{5}$$

Building on Lemma 1, we present the following theorem.

**Theorem 1.** *Given a unique ordered pair $(x, \hat{y})$ and a well-trained network $f(\cdot)$, the gradients for a loss function $J(y_I, \hat{y}), I \in [1, N]$ w.r.t. classes when $y_I$ are $N$ orthogonal one-hot vectors is equivalent to when $y_I$ is a vector of all ones, under the second-order Taylor series approximation.*

*Proof.* Provided in Appendix B.2. □

The proof follows Lemma 1. Theorem 1 states that backpropagating a vector of all ones $(\mathbf{1}_N)$ is equivalent to backpropagating $N$ one-hot vectors with ones at orthogonal positions. This reduces the time complexity from $\mathcal{O}(N)$ to a constant $\mathcal{O}(1)$ since we only require a single pass to backpropagate $\mathbf{1}_N$. Hence, our introspective feature is given by,

$$r_x = \nabla_{W_L} J(\mathbf{1}_N, \hat{y}), r_x \in \Re^{d_{L-1} \times N}, \mathbf{1}_N = 1^{N \times 1} \tag{6}$$

Note the LHS is now $r_x$ instead of $r_I$ from Eq. 5. The final introspective feature is a matrix of the same size as $W_L$ extracted in $\mathcal{O}(1)$ time with a space complexity of $\mathcal{O}(d_{L-1} \times N)$. $r_x$ is vectorized and scaled between $[-1, 1]$ before being used in Sections 4 and 6 as introspective features. This procedure is illustrated in Fig. 3.

# 4 Introspective Network

Once $r_x$ are extracted using Eq. 6, the introspective label $\tilde{y}$ from Fig. 3 is given by $\tilde{y} = \mathcal{H}(r_x)$ where $\mathcal{H}(\cdot)$ is an MLP. In this section, we analyze $\mathcal{H}(\cdot)$. From Fig. 3, $f(\cdot)$ is any existing trained network used to obtain introspective features $r_x$. It is trained to predict the ground truth $y$ given any $x$. Based on the assumption that $\mathcal{H}(r_x) = \mathbb{E}(y|f(x))$ and hence expectation of $y - \mathcal{H}(r_x)$ is 0, the loss function can be decomposed as,

$$\mathbb{E}[(f(x) - y)^2] = \mathbb{E}[(f(x) - \mathcal{H}(r_x))^2)] + \mathbb{E}[(\mathcal{H}(r_x) - y)^2)]. \tag{7}$$

Note that since the goal is to predict $y$ given $x$, $\mathcal{H}(r_x) = \mathbb{E}(y|f(x))$ is a fair assumption to make. Substituting for $f(x)$ in Eq. 7, and using variance decomposition of $y$ onto $f(x)$, we have,

$$\mathbb{E}[(\hat{y} - y)^2] = \text{Var}(\hat{y}) - \text{Var}(\mathcal{H}(r_x)) + \mathbb{E}[(\mathcal{H}(r_x) - y)^2]. \tag{8}$$

This decomposition is adopted from structured calibration techniques. A full derivation is presented in (25). The first term $\text{Var}(\hat{y})$ is the the variance in the prediction from $f(\cdot)$. This term is the precision of $f(\cdot)$ and is low for a well trained network. The third term is the MSE function between the introspective network $\mathcal{H}(\cdot)$ and the ground truth. It is minimized while training the $\mathcal{H}(\cdot)$ network. The second term is the variance of the network $\mathcal{H}(\cdot)$, given features $r_x$. Note that minimizing Eq. 8 can occur by maximizing $\text{Var}(\mathcal{H}(r_x))$. We use a fisher vector interpretation from (17) to analyze $\text{Var}(\mathcal{H}(r_x))$. If $\mathcal{H}(\cdot)$ is a linear layer with parameters $W_\mathcal{H}$, the $\text{Var}(\mathcal{H}(r_x))$ term reduces to $W_\mathcal{H}^T W_\mathcal{H} \times \text{Var}(r_x) \propto \text{Tr}(r_x^T \Sigma^{-1} r_x)$ where $\Sigma$ is the covariance matrix. $\Sigma$ is a gaussian approximation for the shape of the manifold. Generalizing it to a higher dimensional manifold and replacing $\Sigma$ with $F$, we have,

$$\text{Var}(\mathcal{H}(r_x)) = \text{Tr}(r_x^T F^{-1} r_x), \tag{9}$$

$$\text{Var}(\mathcal{H}(r_x)) = \sum_{j=1}^{N} r_j^T F^{-1} r_j. \tag{10}$$

The RHS of Eq. 10 is a sum of fisher vectors taken across all possible labels.

**When do introspective networks provide robustness?** We use Eq. 10 to analyze introspective learning usage. Specifically, we consider two cases: When input $x$ is taken from the training distribution $\mathcal{X}$, and when it is taken from a noisy distribution $\mathcal{X}'$.

1. When a sample $x \in \mathcal{X}$ is provided to a network $f(\cdot)$ trained on $\mathcal{X}$, all $r_j, j \neq \hat{y}$ in Eq. 10 tend to 0. The RHS reduces to $r_{\hat{y}}^T F^{-1} r_{\hat{y}}$. $r_{\hat{y}}$ is a function of $f(x)$ only and hence adds no new information to the framework. The results of $\mathcal{H}(\cdot)$ remain the same as $f(\cdot)$. In other words, given a trained ResNet-18 on CIFAR-10, the results of feed-forward learning will be the same as introspective learning on CIFAR-10 testset.

2. When a new sample $x' \notin \mathcal{X}$ is provided to a network $f(\cdot)$ trained on $\mathcal{X}$, a fisher vector based projection across labels is more descriptive compared to a feed-forward approach. The $N$ gradients in Eq. 10 add new information based on how the network needs to change the manifold shape $F$ to accommodate the introspective gradients. Hence, given a distorted version of CIFAR-10 testset, our proposed introspective learning generalizes with a higher accuracy while providing calibrated outputs from Eq. 8. We empirically illustrate these claims in Section 6. We motivate other applications including active learning, Out-Of-Distribution (OOD) detection, uncertainty estimation, and Image Quality Assessment (IQA) that are dependent on generalizability and calibration in Section 5.

# 5 Related Works for Considered Applications

**Augmentations and Robustness**  The considered $r_x$ features from Eq. 6 can be considered as feature augmentations. Augmentations, including SimCLR (19), Augmix (26), adversarial augmentation (27), and noise augmentations (28) have shown to increase robustness of neural networks. We use introspection on top of non-augmented and augmented (Section 6) networks and show that our proposed two-stage framework increases the robustness to create generalizable and calibrated inferences which aids active learning and out-of-distribution (OOD) detection. The same framework that robustly recognizes images despite noise can also detect noise to make an out-of-distribution detection.

**Confidence and Uncertainty**    The existence of adversarial images ([11]) heuristically decouples the probability of neural network predictions from confidence and uncertainty. A number of works including ([29]) and ([30]) use bayesian formulation to provide uncertainty. However, in downstream tasks like active learning and Out-Of-Distribution (OOD) detection applications, existing state-of-the-art methods utilize softmax probability as confidences. This is because of the simplicity and ease of numerical computation of softmax. In active learning, uncertainty is quantified by the entropy ([31]), least confidence ([31]), or maximum margin ([32]) of predicted logits, or through extracted features in BADGE ([33]), and BALD ([34]). In OOD detection, ([35]) proposes Maximum Softmax Probability (MSP) as a baseline method by creating a threshold function on the softmax output. ([36]) proposes ODIN and improves on MSP by calibrating the network's softmax probability using temperature scaling ([37]). In this paper, we show that the proposed introspective features are better calibrated than their feed-forward counterparts. Hence existing methods in active learning and OOD detection have a superior performance when using $\mathcal{H}(\cdot)$ to make predictions.

**Human Introspection**    We are unaware of any direct application that tests visual human introspection. In its absence, we choose the application of Full-Reference Image Quality Assessment (FR-IQA) to connect machine vision with human vision. The goal in FR-IQA is to objectively estimate the subjective quality of an image. Humans are shown a pristine image along with a distorted image and asked to score the quality of the distorted image ([38]). This requires reflection on the part of the observers. We take an existing algorithm ([39]) and show that introspecting on top of this IQA technique brings its assessed scores closer to human scores.

# 6   Experiments

Across all applications except in Ablation studies in Table 9, we use a 3-layered MLP with sigmoid activations as $\mathcal{H}(\cdot)$. The structure is presented in Appendix C.1. We first define robustness and calibration in the context of this paper.

**Robustness**    In this paper, without loss of consistency with related works, we say that the network trained on distribution $\mathcal{X}$ is robust if it correctly predicts on a shifted distribution $\mathcal{X}'$. The difference in data distributions can be because of data acquisition setups, environmental conditions, distortions among others. We use CIFAR-10 for $\mathcal{X}$ and two distortion datasets - CIFAR-10C ([27]) and CIFAR-10-CURE ([40]) as $\mathcal{X}'$. Generalization is measured through performance accuracy.

**Calibration**    Given a data distribution $x \in \mathcal{X}$, belonging to any of $y \in [1, N]$, a neural network provides two outputs - the decision $\hat{y}$ and the confidence associated with $\hat{y}$, given by $\hat{p}$. Let $p$ be the true probability empirically estimated as $p = \hat{p}_i, \forall i \in [1, M]$. Then calibration is given by ([37]),

$$\mathbb{P}(y = \hat{y} | p = \hat{p}) = p \tag{11}$$

Calibration measures the difference between the confidence levels and the prediction accuracy. To showcase calibration we use the metric of Expected Calibration Error (ECE) as described in ([37]). The network predictions are placed in 10 separate bins based on their prediction confidences. Ideally, the accuracy equals the mid-point of confidence bins. The difference between accuracy and mid-point of bins, across bins is measured by ECE. Lower the ECE, better calibrated is the network.

**Datasets and networks**    CIFAR-10C consists of $950,000$ images whose purpose is to evaluate the robustness of networks trained on original CIFAR-10 trainset. CIFAR-10C perturbs the CIFAR-10 testset using 19 distortions in 5 progressive levels. Hence, there are 95 separate $\mathcal{X}'$ distributions to test on with each $\mathcal{X}'$ consisting of 10000 images. Note that we are not using any distortions or data from CIFAR-10C as a validation split during training. The authors in ([40]) provide realistic distortions that they used to benchmark real-world recognition applications including Amazon Rekognition and Microsoft Azure. We use these distortions to perturb the test set of CIFAR-10. There are 6 distortions, each with 5 progressive levels. Of these 6 distortions - Salt and Pepper, Over Exposure, and Under Exposure noises are new compared to CIFAR-10C. We train four ResNet architectures - ResNet-18, 34, 50, and 101 ([4]). All four ResNets are evaluated as sensing networks $f(\cdot)$. The training procedure and hyperparameters are presented in Appendix C.1.

**Testing on CIFAR-10 testset**    The trained networks are tested on CIFAR-10 testset with accuracies $91.02\%, 93.01\%, 93.09\%$, and $93.11\%$ respectively. Next we extract $r_x$ on all training and testing images in CIFAR-10. $\mathcal{H}(\cdot)$ is trained using $r_x$ from the trainset using the same procedure as $f(\cdot)$. When tested on $r_x$ of the testset, the accuracy for ResNets-18,34,50,101 is $90.93\%, 92.92\%, 93.17\%$,

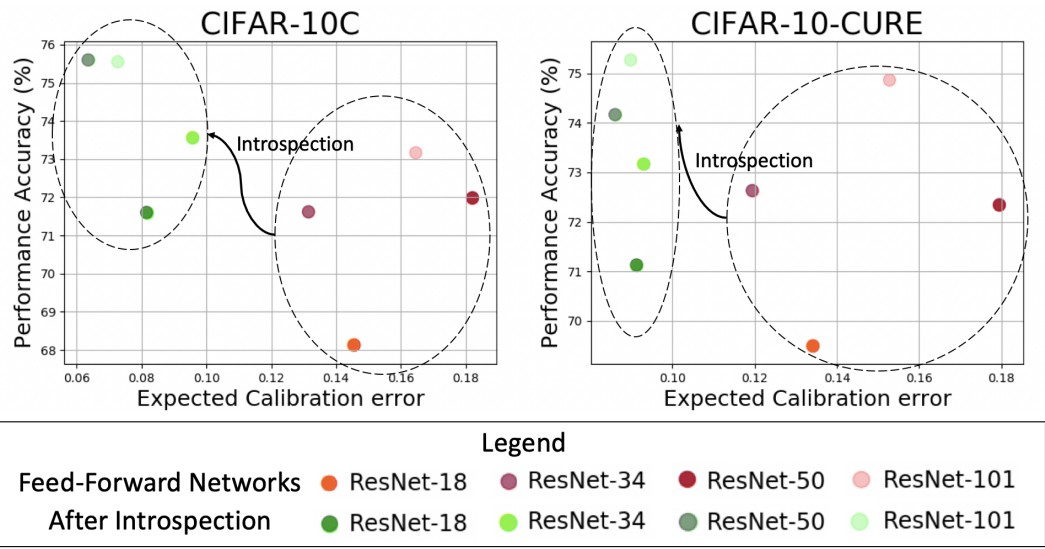

Figure 4: Scatter plot with performance accuracy vs expected calibration error. Ideally, networks are in top left. Introspectivity increases performance accuracy while decreasing calibration error.

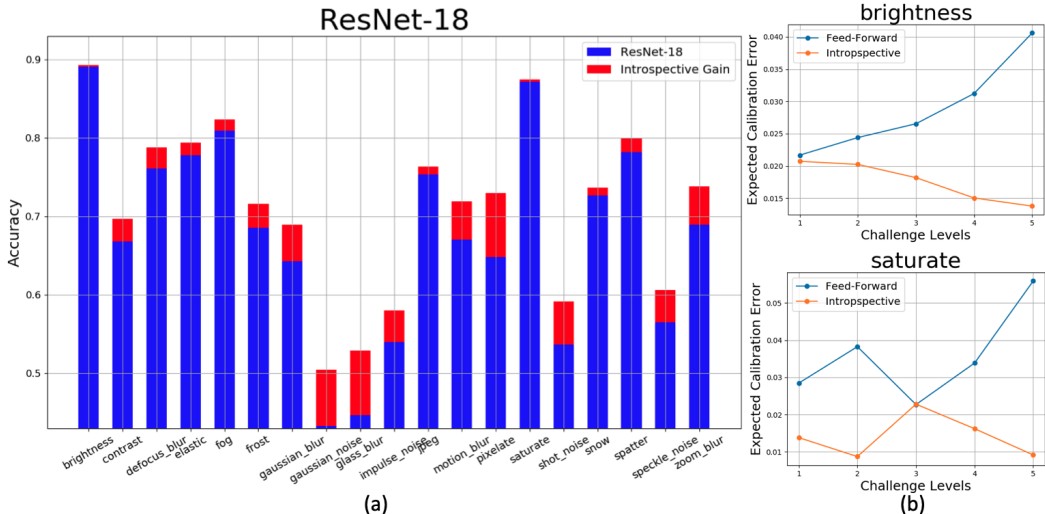

Figure 5: (a) ResNet-18 on CIFAR-10C. (b) Expected calibration error across 5 challenge levels in brightness and saturate distortions. Note that both these distortions do not affect the performance of the network and their feed-forward accuracy is high. The improvement in accuracy is statistically insignificant. However, introspection decreases the ECE across challenge levels.

and $93.03\%$. Note that this is similar to the feed-forward results. The average ECE of all feed-forward and introspective networks is $0.04$. Hence, when the test distribution is the same as training distribution there is no change in performance.

**Testing on CIFAR-10C and CIFAR-10-CURE** The results of all networks averaged across distortions in both the datasets are shown in Fig. 4. Note that in each case, there is a shift leftward and upward indicating that the performance improves while the calibration error decreases. In the larger CIFAR-10C dataset, the introspective ResNet-18 performs similar to ResNets-34 and 50 in terms of accuracy while beating them both in calibration. A more fine-grained analysis is shown in Fig. 5 for ResNet-18. The blue bars in Fig. 5(a) represent the feed-forward accuracy. The red bars are the introspective accuracy gains over the feed-forward accuracy. Among 7 of the 19 distortions, the accuracy gains are over $5\%$. In Appendix C.2.1, we see that the gains are higher when the distortions are higher. Introspection performs well on blur-like distortions while struggling with distortions that disrupt the lower level characteristics of the image like brightness, contrast, and saturate. This can be attributed to the fact that $r_x$ are derived from the last layer of $f(\cdot)$ and are missing low-level

statistics that are filtered out by network in the initial layers. However, in Fig. 5b), we show ECE for brightness and saturate distortions across all $5$ distortion levels - higher the level, more is the distortion affecting $\mathcal{X}'$. It can be seen that while the ECE for feed-forward networks increases across levels, the ECE for introspective networks decrease. Hence, even when there are no accuracy gains to be had, introspection helps in calibration.

**Plug-in results of Introspection** Note that there are a number of techniques proposed to alleviate a neural network's robustness challenges against distortions. The authors in (28) show that finetuning VGG-16 using blurry training images increases the performance of classification under blurry conditions. (41) propose utilizing distorted virtual images to boost performance accuracy. The authors in (27) use adversarial images to augment the training data. All these works require knowledge of distortion or large amounts of new data during training. Our proposed method can infer introspectively on top of any existing $f(\cdot)$ enhanced using existing methods. In Table 1, we show results of introspection as a *plug-in* approach on top of existing techniques. In all methods, introspection outperforms its feed-forward counterpart. While introspecting on top of Augmix, trained on WideResNet (26) provides insignificant recognition accuracy gains, introspection reduces ECE of Augmix network by $43.33\%$. In Appendix C.2, we show performance on top of (28) and (27) of $6.8\%$ on Level 5 distortions. In Appendix C.3, we analyze SimCLR and show that introspecting on the self supervised features increases its CIFAR-10C performance by about $6\%$ on ResNet-101. A number of ablation studies including analysis of structure of $\mathcal{H}(\cdot)$, loss functions, distortion levels on performance accuracy and ECE are shown in Appendix C.4. Moreover, we examine introspection when $\mathcal{X}'$ is domain shifted data from Office (42) dataset in Appendix C.6.1.

Table 1: Introspecting on top of existing robustness techniques.

| METHODS | | ACCURACY |
|---|---|---|
| RESNET-18 | FEED-FORWARD | 67.89% |
| | INTROSPECTIVE | **71.4%** |
| DENOISING | FEED-FORWARD | 65.02% |
| | INTROSPECTIVE | **68.86%** |
| ADVERSARIAL TRAIN (27) | FEED-FORWARD | 68.02% |
| | INTROSPECTIVE | **70.86%** |
| SIMCLR (19) | FEED-FORWARD | 70.28% |
| | INTROSPECTIVE | **73.32%** |
| AUGMENT NOISE (28) | FEED-FORWARD | 76.86% |
| | INTROSPECTIVE | **77.98%** |
| AUGMIX (26) | FEED-FORWARD | 89.85% |
| | INTROSPECTIVE | **89.89%** |

**Downstream Applications** We consider four downstream applications: Active Learning, Out-of-Distribution detection, Uncertainty estimation, and Image Quality Assessment (IQA) to demonstrate the validity of introspection. Similar to recognition experiments, in all considered applications there is a distributional difference between train and testset. We show results that conform to Eq. 10. We show statistically significant introspective IQA results in Appendix D.3 and Table 13, and uncertainty estimation results in Appendix D.4.

**Active Learning** The goal in active learning is to decrease the test error in a model by choosing the *best* samples from a large pool of unlabeled data to annotate and train the model. A number of strategies are proposed to query the *best* samples. A full review of active learning and query strategies are given in (43). Existing active learning strategies define *best* samples to annotate as those samples that the model is either most uncertain about. This uncertainty is quantified by either entropy (31), least confidence (31), maximum margin (32), or through extracted features in BADGE (33), and BALD (34). We show the results of ResNet-18 and 34 architecture in Table 2. Implementations of all query strategies in Table 2 are taken from the codebase of (33) and reported as Feed-Forward results. Note that the query strategies act on $f(\cdot)$ to sample images at every round. Instead of sampling on $f(\cdot)$, all query strategies sample using $\mathcal{H}(\cdot)$ in the Introspective results. The training, testing, and all strategies are the same as Feed-Forward from (33). Doing so we find similar results as recognition - on the original testset the active learning results are the same while there is a gain across strategies on Gaussian noise testset from CIFAR-10C. Note that the results shown are averaged over 20 rounds with a query batch size of a 1000 and initial random choice - which were kept same for $f(\cdot)$ and $\mathcal{H}(\cdot)$ - of 100. Further details, plots, and variances are shown in Appendix D.1.

**Out-of-distribution Detection** The goal of Out-Of-Distribution (OOD) detection is to detect those samples that are drawn from a distribution $\mathcal{X}' \neq \mathcal{X}$ given a fully trained $f(\cdot)$. A number of techniques are proposed to detect out-of-distribution samples. The authors in (35) propose Maximum Softmax Probability (MSP) as a baseline method by creating a threshold function on the softmax output. The authors in (36) propose ODIN and improved on MSP by calibrating the network's softmax probability using temperature scaling (37). In this paper, we illustrate that applying existing methods when applied on $\mathcal{H}(\cdot)$, their detection performance is greater than if they were applied on the feed-forward

Table 2: Recognition accuracy of Active Learning strategies.

| Methods | Architecture | Original Testset | | Gaussian Noise | |
|---|---|---|---|---|---|
| | | R-18 | R-34 | R-18 | R-34 |
| Entropy [31] | Feed-Forward | 0.365 | 0.358 | 0.244 | 0.249 |
| | Introspective | 0.365 | 0.359 | **0.258** | **0.255** |
| Least [31] | Feed-Forward | 0.371 | 0.359 | 0.252 | 0.25 |
| | Introspective | 0.373 | 0.362 | **0.264** | **0.26** |
| Margin [32] | Feed-Forward | 0.38 | 0.369 | 0.251 | 0.253 |
| | Introspective | 0.381 | 0.373 | **0.265** | **0.263** |
| BALD [34] | Feed-Forward | 0.393 | 0.368 | 0.26 | 0.253 |
| | Introspective | 0.396 | 0.375 | **0.273** | **0.263** |
| BADGE [33] | Feed-Forward | 0.388 | 0.37 | 0.25 | 0.247 |
| | Introspective | 0.39 | 0.37 | **0.265** | **0.260** |

Table 3: Out-of-distribution Detection of existing techniques compared between feed-forward and introspective networks.

| Methods | OOD Datasets | FPR (95% at TPR) ↓ | Detection Error ↓ | AUROC ↑ |
|---|---|---|---|---|
| | | Feed-Forward/Introspective | | |
| MSP [35] | Textures | 58.74/**19.66** | 18.04/**7.49** | 88.56/**97.79** |
| | SVHN | 61.41/**51.27** | 16.92/**15.67** | 89.39/**91.2** |
| | Places365 | 58.04/**54.43** | 17.01/**15.07** | 89.39/**91.3** |
| | LSUN-C | **27.95**/27.5 | **9.42**/10.29 | **96.07**/95.73 |
| ODIN [36] | Textures | 52.3/**9.31** | 22.17/**6.12** | 84.91/**91.9** |
| | SVHN | 66.81/**48.52** | 23.51/**15.86** | 83.52/**91.07** |
| | Places365 | **42.21**/51.87 | 16.23/**15.71** | **91.06**/90.95 |
| | LSUN-C | **6.59**/23.66 | **5.54**/10.2 | **98.74**/ 95.87 |

$f(\cdot)$. The code for OOD detection techniques are taken from [44] along with all hyperparameters and the training regimen for their reported DenseNet [5] architecture. The temperature scaling coefficient for ODIN is set to 1000. Note that we do not use temperature scaling on $\mathcal{H}$, to illustrate the effectiveness of our method. We use three established metrics to evaluate OOD detection - False Positive Rate (FPR) at 95% True Positive Rate (TPR), Detection error, and AUROC. Ideally, AUROC values for a given method is high while the other two metrics are low. We use CIFAR-10 as our in-distribution dataset and use four OOD datasets - SVHN [45], Describable Textures Dataset [46], Places 365 [47], and LSUN [48]. The results are presented in Table 3. Note that among the four datasets, textures and SVHN are *more* out-of-distribution from CIFAR-10 than the natural image datasets of Places365 and LSUN. The results of the introspective network is highest on Textures DTD dataset and gets progressively worse among the natural image datasets. Further analysis on networks and methods, along with their training regimen is provided in Appendix D.2.

## 7  Discussion and Conclusion

**Limitations and future work**   The paper illustrates the benefits of utilizing the change in model parameters as a measure of model introspection. In Section 3.2, we accelerate the time complexity to $\mathcal{O}(1)$. However, the space complexity is still dependent on $N$. The paper uses an MLP for $\mathcal{H}(\cdot)$ and constructs $r_x$ by vectorizing extracted gradients. For datasets with large $N$, usage of $r_x$ as a vector is prohibitive. Hence, a required future work is to provide a method of combining all $N$ gradients without vectorization. Also, our implementation uses serial gradient extraction across images. This is non-ideal since the available GPU resources are not fully utilized. A parallel implementation with per-sample gradient extraction [49] is a pertinent acceleration technique for the future.

**Broader and Societal Impact**   In his seminal book in 2011 [50], Daniel Kahneman outlines two systems of thought and reasoning in humans - a fast and instinctive 'system 1' that heuristically associates sensed patterns followed by a more deliberate and slower 'system 2' that examines and analyzes the data in context of intrinsic biases. Our framework derives its intuition based on these two systems of reasoning. The introspective explanations can serve to examine the intrinsic notions and biases that a network uses to categorize data. Note that the network $\mathcal{H}(\cdot)$ obtains its introspective answers through $f(\cdot)$. Hence, similar to the 'system 2' reasoning in humans, any internal bias present in $f(\cdot)$ only gets strengthened in $\mathcal{H}(\cdot)$ through confirmation bias. The framework will benefit from a human intervention between $f(\cdot)$ and $\mathcal{H}(\cdot)$ in sensitive applications. One way would be to ask counterfactual questions by providing an established counterfactual and asking the network to reflect based on that. While the introspective framework will remain the same, the features will change.

**Conclusion**   We introduce the concept of introspection in neural networks as two separate stages in a network's decision process - the first is making a quick assessment based on sensed patterns in data and the second is reflecting on that assessment based on all possible decisions that could have been taken and making a final decision based on this reflection. We show that doing so increases the generalization performance of neural networks as measured against distributionally shifted data while reducing the calibration error of neural networks. Existing state-of-the-art methods in downstream tasks like active learning and out-of-distribution detection perform better in an introspective setting compared to a feed-forward setting especially when the distributional difference is high.

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
