# A   Appendix : Introspection, Reasoning, and Explanations

Introspection was formalized by (51) as a field in psychology to understand the concepts of memory, feeling, and volition (52). The primary focus of introspection is in reflecting on oneself through directed questions. While the directed questions are an open field of study in psychology, we use reasoning as a means of questions in this paper. Specifically, abductive reasoning. Abductive reasoning was introduced by the philosopher Charles Sanders Peirce (53), who saw abduction as a reasoning process from effect to cause (54). An abductive reasoning framework creates a hypothesis and tests its validity without considering the cause. From the perspective of introspection, a hypothesis can be considered as an answer to one of the three following questions: a correlation *'Why P?'* question, a counterfactual *'What if?'* question, and a contrastive *'Why P, rather than Q?'* question. Here $P$ is the prediction and $Q$ is any contrast class. Both the correlation and counterfactual questions require active interventions for answers. These questions try to assess the causality of some endogenous or exogenous variable and require interventions that are long, complex, and sometimes incomplete (55). However, introspection is the assessment of ones own notions rather than an external variable. Hence, a contrastive question of the form *'Why P, rather than Q?'* lends itself as the directed question for introspection. Here $Q$ is the introspective class. It has the additional advantage that the network $f(\cdot)$ serves as the knowledge base of notions. All reflection images from 1, Fig. 6, and Fig. 2 are contrastive. We describe the generation process of these *post-hoc* explanations.

**Introspective Feature Visualization**   We modify Grad-CAM (6) to visualize $r_j$ from Eq. 2. Grad-CAM visually justifies the decision made by $f(\cdot)$ by highlighting features that lead to $\hat{y}$. It does so by backpropagating the logit associated with the prediction, $\hat{y}$. The resulting gradients at every feature map are global average pooled and used as importance scores. The importance scores multiply the activations of the final convolutional layer and the resultant map is the Grad-CAM visualization. Hence, gradients highlight the activation areas that maximally lead to the prediction $\hat{y}$. In Fig. 1, given a spoonbill image $x$ and a ImageNet-pretrained (23) VGG-16 network, the sensing visualization shown is Grad-CAM. Grad-CAM indicates that the pink and round body, and straight beak are the reasons for the decision. Instead of backpropagating the $\hat{y}$ logit, we backpropagate $J(y_I, \hat{y})$ in the Grad-CAM framework. The gradients represent introspective features and are used as importance scores. It can be seen that they visually highlight the explanations to *'Why $\hat{y}$, rather than $y_I$'*. In Fig. 1, the network highlights the neck of the spoonbill to indicate that since an S-shaped neck is not observed, $x$ cannot be a flamingo. Similarly, the body of the spoonbill is highlighted when asked why $x$ is not a crane since cranes have white feathers while spoonbills are pink. Two more examples are shown in Fig. 6. In the first row, a VGG-16 architecture is trained on Stanford Cars dataset (56). Given a Bugatti convertible image, Grad-CAM highlights the bonnet as the classifying factor. An introspective question of why it cannot be a bugatti coupe is answered by highlighting the open top of the convertible. The entire car is highlighted to differentiate the bugatti convertible from a Volvo. In the second row, we explore visual explanations in computed seismic images using LANDMASS dataset (57). A ResNet-18 architecture using the procedure from (58) is trained. The dataset has four geological features as classes - faults, salt domes, horizons, and chaotic regions. Given a fault image in Fig. 6, Grad-CAM highlights the regions where the faults are clearly visible as fractures between rocks. However, these regions resemble salt domes as shown in the representative image. The introspective answer of why $x$ is not predicted as a salt dome tracks a fault instead of highlighting a general region that also resembles a salt dome. Note that no representative images are required to obtain introspective visualizations. The gradients introspect based on notions of classes in network parameters.

**Biological plausibility of introspection in recognition**   Recognition is fast and mostly a feed-forward process. However, when there is uncertainty involved - either due to distributional shift or noise - we tend to reason about our decisions. Human visual system detects salient portions of an image and attends to them in a feed-forward process. An alternative perspective to this is expectancy-mismatch - the idea that HVS attends to those features that deviate from expectations (15). We simulate this via introspection. By asking *'Why P, rather than Q?'*, we ask the network to examine its expectations and describe the mismatches. This is seen as the lack of S-shaped neck in spoonbill in Fig. 1. By interpreting introspective features as hypotheses that answer contrastive questions, we convert the biologically feed-forward process into a reflection process. Moreover, our results also support this - when train and testsets are from the same distribution, there is no change in results. However, when there is a distributional difference, we notice the gains for introspection - on

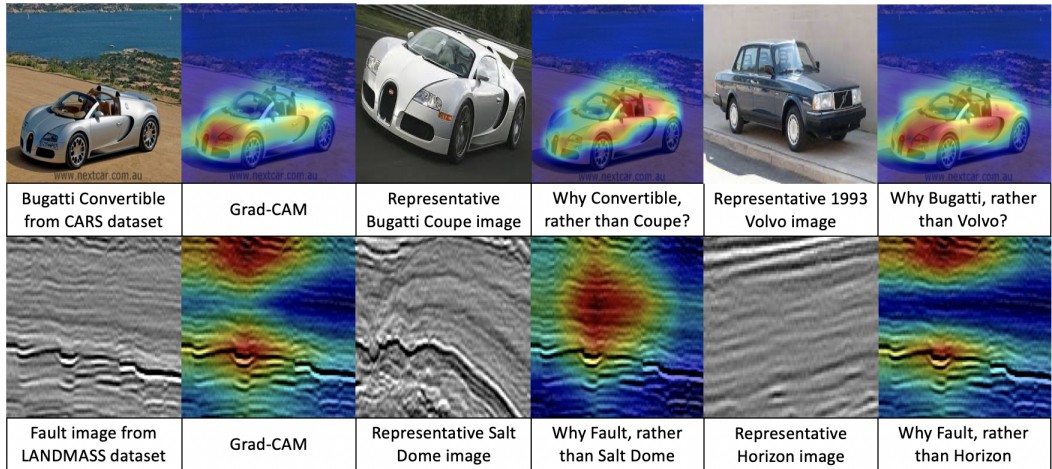

| Bugatti Convertible from CARS dataset | Grad-CAM | Representative Bugatti Coupe image | Why Convertible, rather than Coupe? | Representative 1993 Volvo image | Why Bugatti, rather than Volvo? |
| Fault image from LANDMASS dataset | Grad-CAM | Representative Salt Dome image | Why Fault, rather than Salt Dome | Representative Horizon image | Why Fault, rather than Horizon |

Figure 6: Introspective feature visualizations. The images in the leftmost column are the input $x$. The representative images are for illustrative purposes and are not used to extract features.

CIFAR-10C, CIFAR-10-CURE, active learning, OOD and IQA experiments. Moreover, higher the distributional difference, larger is the introspective gain as shown in Fig. 8.

**A broader view on Introspection** In this paper, we limit introspection as answers to *'Why P, rather than Q?'* questions. We limit $P$ to be the predictions made by networks. Hence, we are left with $N$ questions to answer. However, creative abduction calls for asking questions of the form *'Why P, rather than Q1 and Q2?'*. Such $Qs$ can extend to all $N$ classes. Hence, introspective labels can be the powerset of all one-hot labels - $2^N$. Moreover, the prediction itself can be made to change by intervening within data leading to questions of the form *'Why Q, rather than P?'*. These include counterfactual questions of the form *'What if?'* when considered from the output perspective. Hence, for $N$ classes, there can be $N \times 2^N$ introspective questions. Hence, the proposed features are only one possible feature set when considering introspection. However, we posit that all these features are a function of the data and the model, thereby making gradients an essential feature set while considering introspection and this paper provides intuitions as to their applicability.

# B    Appendix : Proofs

## B.1    Proof for Lemma 1

We start by assuming $J(\cdot)$ is a cross-entropy loss. $J(y_I, \hat{y}), I \in [1, N]$ can also be written as,

$$J(y_I, \hat{y}) = -y_{\hat{y}} + \log \sum_{j=1}^{N} e^{y_j}, \text{ where } \hat{y} = f(x), \hat{y} \in \Re^{N \times 1}. \tag{12}$$

This definition is used in PyTorch to implement cross entropy. Here we assume that the predicted logit, i.e, the argument of the max value in the logits $\hat{y}$ is $y_{\hat{y}}$. While training, $y_{\hat{y}}$ is the true label. In this paper, we backpropagate any trained class $I$, as an introspective class. Hence, Eq. 12 can be rewritten as,

$$J(y_I, \hat{y}) = -y_I + \log \sum_{j=1}^{N} e^{y_j}, \text{ where } \hat{y} = f(x), \hat{y} \in \Re^{N \times 1}. \tag{13}$$

Approximating the exponent within the summation with its second order Taylor series expansion, we have,

$$J(y_I, \hat{y}) = -y_I + \log \sum_{j=1}^{N} \left( 1 + y_j + \frac{y_j^2}{2} \right). \tag{14}$$

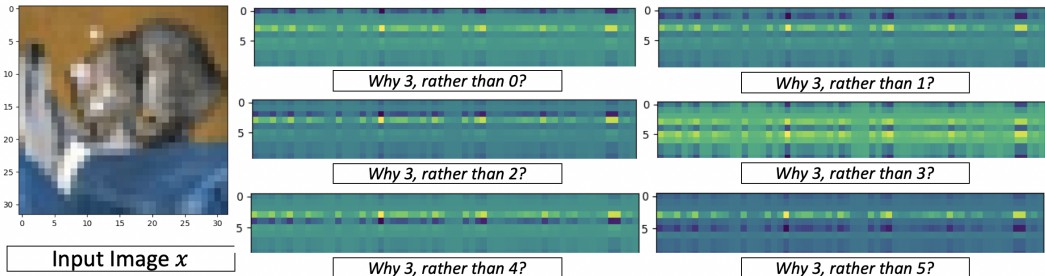

Figure 7: For the input image on the left, the $\nabla_{W_L} J(y_I, 3)$ are shown on the right. Each image is a visualization of the $64 \times 10$ gradient matrix.

Note that for a well trained network $f(\cdot)$, the logits of all but the predicted class are negligible. As noted before, the predicted logit is $y_{\hat{y}}$. Taking $y_{\hat{y}}$ within the summation as common,

$$J(y_I, \hat{y}) = -y_I + \log \sum_{j=1}^{N} y_{\hat{y}} \left( \frac{1}{y_{\hat{y}}} + \frac{y_j}{y_{\hat{y}}} + \frac{y_j^2}{2y_{\hat{y}}} \right). \tag{15}$$

Taking out $y_{\hat{y}}$ from within the summation since it is a constant and independent of summation variable $j$,

$$J(y_I, \hat{y}) = -y_I + \log \left( y_{\hat{y}} \sum_{j=1}^{N} \left( \frac{1}{y_{\hat{y}}} + \frac{y_j}{y_{\hat{y}}} + \frac{y_j^2}{2y_{\hat{y}}} \right) \right). \tag{16}$$

Using product rule of logarithms,

$$J(y_I, \hat{y}) = -y_I + \log(y_{\hat{y}}) + \log \sum_{j=1}^{N} \left( \frac{1}{y_{\hat{y}}} + \frac{y_j}{y_{\hat{y}}} + \frac{y_j^2}{2y_{\hat{y}}} \right). \tag{17}$$

For a well trained network, all logits except $y_{\hat{y}}$ are negligible. Also for a well trained network, $y_{\hat{y}}$ is large. Hence the third term on the RHS within the summation reduces to $1 + \frac{y_{\hat{y}}}{2}$. Note that in the second term of the RHS, $y_{\hat{y}} = c$ is a constant for any deterministic trained network even when there are small changes in the values of weights $W$. Substituting,

$$J(y_I, \hat{y}) = -y_I + \log(c) + \log \left( 1 + \frac{y_{\hat{y}}}{2} \right). \tag{18}$$

The quantity in Eq. 18 is differentiated, hence nulling the effect of constant $\log(c)$. Hence, we can obtain $\nabla_W J(y_j, \hat{y})$ as a function of two logits, $y_I$ and $y_{\hat{y}}$ given by,

$$\nabla_W J(y_I, \hat{y}) = -\nabla_W y_I + \nabla_W \log \left( 1 + \frac{y_{\hat{y}}}{2} \right). \tag{19}$$

$y_I$ is a one-hot vector of dimensionality $N \times 1$ while $\nabla_W$ is a $d_{L-1} \times N$ matrix. The product extracts only the $I^{th}$ filter in the $W$ matrix in gradient calculations. Following the above logic for $y_{\hat{y}}$, we have,

$$\nabla_W J(y_I, \hat{y}) = -\nabla_{W,I} y_I + \nabla_{W,y_{\hat{y}}} g(y_{\hat{y}}), \tag{20}$$

where $g(\cdot)$ is some function of $y_{\hat{y}}$. Hence the gradient $r_I = \nabla_W J(y_I, \hat{y})$ lies in the span of the filter gradients of $W_I$ and $W_{\hat{y}}$, making $r_I$ orthogonal to all other filter gradient pairs. Hence proven.

Hence, for $N$ introspective features in 5, the space complexity of $r_x$ which is a concatenation of $N$ separate $r_i$, reduces from $\mathcal{O}(d_{L-1} \times N^2)$ to $\mathcal{O}(d_{L-1} \times N)$.

Similar to Fig. 2 that was presented on a well trained network on MNIST dataset, we show the sparsity analysis on CIFAR-10 data in Fig. 7. The sparse nature of gradients is still observed in Fig. 7 but it is not as prevalent as the gradients from Fig. 2. This is because of the assumption of the well trained network in Lemma 1. This assumption allows for Eq. 18 where we assume only the predicted logit and its closest logit are non-zero. However, in Table 9, we show that the approximation does not alter the empirical results since the excess non-zero logits tend to store redundant information across filters. This is also observable in Fig. 7.

Table 4: Structure of $\mathcal{H}(\cdot)$ and accuracies on CIFAR-10C as reported in the paper.

| (Training Domain)$f(\cdot)$ | Part 1: Structure of $\mathcal{H}(\cdot)$ - All layers separated by sigmoid | Accuracy (%) |
|---|---|---|
| (CIFAR-10) R-18,34 | $640 \times 300 - 300 \times 100 - 100 \times 10$ | 71.4, 73.36 |
| (CIFAR-10) R-50, 101 | $2560 \times 300 - 300 \times 100 - 100 \times 10$ | 75.2, 75.47 |
| (Webcam) R-18,34 | $1984 \times 31$ | - |
| (Webcam) R-50,101 | $7936 \times 31$ | - |
| (Amazon) R-18,34 | $1984 \times 1000 - 1000 \times 100 - 100 \times 31$ | - |
| (Amazon) R-50,101 | $7936 \times 3000 - 3000 \times 500 - 500 \times 31$ | - |
| (DSLR) R-18,34 | $1984 \times 1000 - 1000 \times 100 - 100 \times 31$ | - |
| (VisDA) R-18 | $768 \times 300 - 300 \times 100 - 100 \times 12$ | - |

## B.2 Proof for Theorem 1

The proof for Theorem 1 follows from Lemma 1. For any given data $x$, there are $N$ possible introspections and hence $N$ possible reflections. The LHS in Eq. 20 is summed across $N$ losses. Since $y_j, j \in [1, N]$ are one-hot vectors, they are orthogonal and the first term in RHS is an addition across $j$. The second term in RHS is independent of $j$. Representing this in equation form, we have,

$$\sum_{j=1}^{N} \nabla_W J(y_j, \hat{y}) = -\sum_{j=1}^{N} \nabla_{W,j} y_j + N \times \nabla_{W, y_{\hat{y}}} g(y_{\hat{y}}). \tag{21}$$

The first term is added $N$ times for $N$ orthogonal $y_I$. Hence, the first term reduces to a sum of all gradients of $j^{th}$ filters when backpropagating $y_j$. Removing the summation and replacing $y_j = \mathbf{1}_N$ or a vector of all ones in the LHS, we still have the same RHS given by,

$$\nabla_W J(\mathbf{1}_N, \hat{y}) = -\sum_{j=1}^{N} \nabla_{W,j} y_j + N \times \nabla_{W, y_{\hat{y}}} g(y_{\hat{y}}). \tag{22}$$

Equating the LHS from Eq. 21 and Eq. 22, we have the proof.

## B.3 Tradeoff in Eq. 8

Eq. 8 suggests a trade-off between minimizing $\mathbb{E}[(\mathcal{H}(r_x) - y)^2]$, which is the cost function for training $\mathcal{H}(\cdot)$, and the variance of the network $\mathcal{H}(\cdot)$. Ideally, an optimal point exists that optimally minimizes the cost function of $\mathcal{H}(\cdot)$ while maximizing its variance. This also prevents decomposing $\mathcal{H}(\cdot)$ into $\mathcal{H}_1(\cdot)$ and $\mathcal{H}_2(\cdot)$ that further introspect on $\mathcal{H}(\cdot)$. In this paper, we create a single introspective network $\mathcal{H}(\cdot)$. Hence, we do not comment further on the practical nature of the trade-off or perpetual introspection. It is currently beyond the scope of this work. In all experiments, we train $\mathcal{H}(\cdot)$ as any other network feed-forward network - by minimizing an empirical loss function given the ground truth.

## B.4 Fisher Vector Interpretation

We make two claims before Eq. 10 both of which are well established. These include :

- **Variance of a linear function** For a linear function $y = W \times x + b$, the variance of $y$ is given by $\text{Var}(Wx + b) = W^2 \text{Var}(x)$ if $\text{Var}(W) = 0$.

- **Variance of a linear function when $W$ is estimated by gradient descent** Ignoring the bias $b$, and taking $y = Wx = x^T \Sigma^{-1} x^T (xW)$, we have $\text{Var}(Wx) = \sigma^2 \text{Tr}(x^T \Sigma^{-1} x)$.

Both these results lead to Eq. 9. Since $r_x \in \Re^{d_{L-1} \times N}$, the trace of the matrix given by $\text{Tr}(r_x^T F^{-1} r_x)$, is a sum of projections on individual weight gradients given by $\sum_{j=1}^{N} r_j^T F^{-1} r_j$ in the Fisher sense.

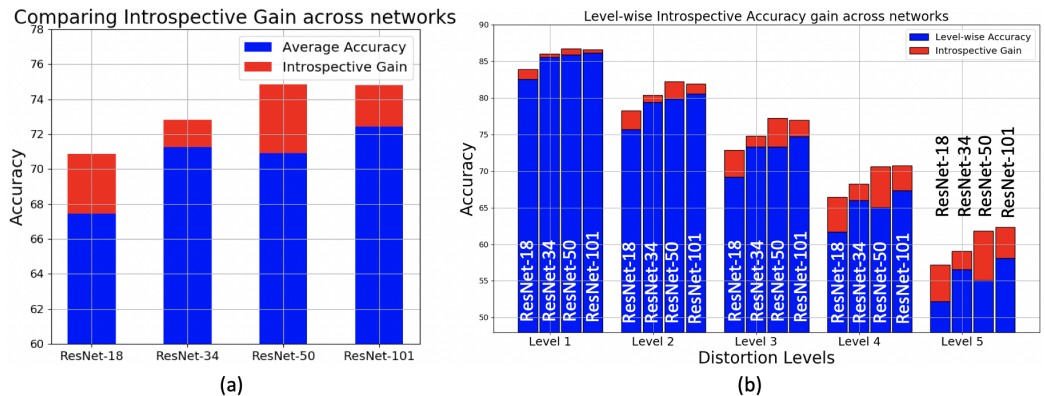

Figure 8: Introspective performance gains over Feed-Forward networks of a) ResNets-18,34,50,101, b) Level-wise averaged results across ResNets-18,34,50,101

# C  Appendix : Additional Results on Recognition and Calibration

## C.1  Structure of $\mathcal{H}(\cdot)$ and training details

In this section, we provide the structure of the proposed $\mathcal{H}(\cdot)$ architecture. Note that, from Eq. 1, the $y_{feat}$ in feed-forward learning are processed through a linear layer. We process the introspective features $r_x$ through an MLP $\mathcal{H}(\cdot)$, whose parameter structure is given in Table 4. Hence, we follow the same workflow as feed-forward networks in introspective learning. The feed-forward features $f_{L-1}(x)$ are passed through the last linear layer in $f(\cdot)$ to obtain the prediction $\hat{y}$. The introspective features are passed through an MLP to obtain the prediction $\tilde{y}$. The exact training procedure for $\mathcal{H}(\cdot)$ is presented below.

**Training $f(\cdot)$ and Hyperparameters**   We train four ResNet architectures - ResNet-18, 34, 50, and 101 (4). Note that we are not using any known techniques that promote either generalization (training on noisy data (28)) or calibration (Temperature scaling (37)). The networks are trained from scratch on CIFAR-10 dataset which consists of 50000 training images with 10 classes. The networks are trained for 200 epochs using SGD optimizer with momentum $= 0.9$ and weight decay $= 5e - 4$. The learning rate starts at $0.1$ and is changed as $0.02, 0.004, 0.0008$ after epochs $60, 120$, and $160$ respectively. PyTorch in-built Random Horizontal Flip and standard CIFAR-10 normalization is used as preprocessing transforms.

**Training $\mathcal{H}(\cdot)$**   The structures of all MLPs are shown in Table 4. ResNet-18,34 trained on CIFAR-10 provide $r_x$ of dimensionality $640 \times 1$. This is fed into $\mathcal{H}(\cdot)$ which is trained to produce a $10 \times 1$ output. Note that $r_x$ from ResNet-50,101 are of dimensionality $2560 \times 1$ - due to larger dimension of $f_{L-1}()$. All MLPs are trained similar to $f(\cdot)$ - for 200 epochs, SGD optimizer, momentum $= 0.9$, weight decay $= 5e^{-3}$, learning rates of $0.1, 0.02, 0.004, 0.0008$ in epochs $1 - 60, 61 - 120, 121 - 160, 161 - 200$ respectively. For the larger 5-layered ResNet-50,101 networks in Table 9, dropout with $0.1$ is used and the weight decay is reduced to $5e^{-4}$.

## C.2  Introspective Accuracy Gain and Calibration Error Studies

In this section, we present additional recognition and calibration results. In Fig. 5a), we showed distortion-wise accuracy and the introspective gain for ResNet-18. In this section, we present level-wise and network-wise accuracies for all four considered ResNet architectures. We show that an introspective ResNet-18 matches a Feed-Forward ResNet-50 in terms of recognition performance. We then compare the results of ResNet-18 against existing techniques that promote robustness. We show that introspection is a plug-in approach that acts on top of existing methods and provides gain. We do the same for calibration experiments on CIFAR-10C where we provide level-wise distortion-wise graphs for Expected Calibration Error (ECE) similar to Fig. 5b).

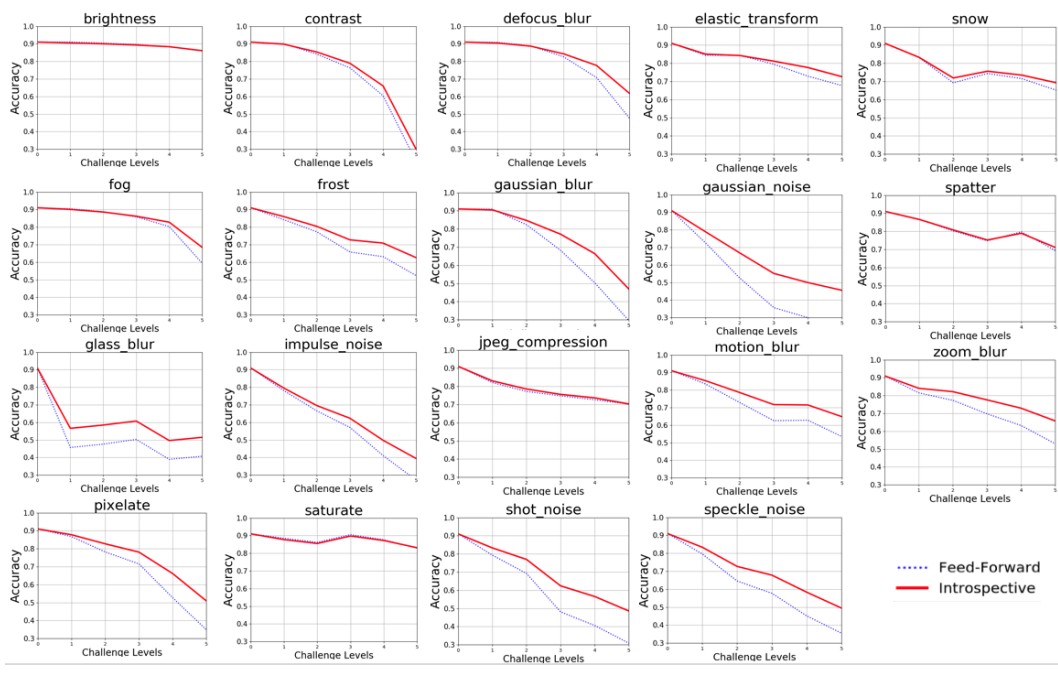

Figure 9: Introspective performance gains over Feed-Forward Resnet-18 across distortions and levels

### C.2.1 Level-wise Recognition on CIFAR-10C

In Fig. 8b), the introspective performance gains for the four networks are categorized based on the distortion levels. All 19 categories of distortion on CIFAR-10C are averaged for each level and their respective feed-forward accuracy and introspective gains are shown. Note that the levels are progressively more distorted. Hence, level 1 distribution $\mathcal{X}'$ is similar to the training distribution $\mathcal{X}$ when compared to level 5 distributions. As the distortion level increases, the introspective gains also increase. This is similar to the results from Section D. In both active learning and OOD applications as $\mathcal{X}'$ deviates from $\mathcal{X}$, introspection performs better. In Fig. 8a), we show the distortion-wise and level-wise increase for each network. Note that, an Introspective ResNet-18 performs similarly to a Feed-Forward ResNet-50. From (4), the number of parameters in a ResNet-18 model ($1.8 \times 10^9$) are less than half the parameters in a ResNet-50 model ($3.8 \times 10^9$). However by adding an introspective model $\mathcal{H}(\cdot)$ with $2.23 \times 10^5$ parameters to a feed-forward ResNet-18 model, we can obtain the same accuracy as a ResNet-50 model. This is in addition to the calibration gains provided by introspection.

### C.2.2 Distortion-wise and Level-wise Recognition on CIFAR-10C

In Fig. 9, the introspective accuracy performance for Resnet-18 across 19 distortions and 5 distortion levels is shown. Note that CIFAR-10C consists of 950,000 test images. The 4% increase in performance translates to around 35,000 more images correctly classified over its feed-forward counterpart. These gains are especially visible among Level 5 distortions.

Table 5: Introspecting on top of existing robustness techniques.

| Methods | Accuracy |
|---|---|
| ResNet-18 | 67.89% |
| Denoising | 65.02% |
| Adversarial Train (27) | 68.02% |
| SimCLR (19) | 70.28% |
| Augment Noise (28) | 76.86% |
| Augmix (26) | 89.85% |
| ResNet-18 + Introspection | 71.4% |
| Denoising + Introspection | 68.86% |
| Adversarial + Introspection | 70.86% |
| SimCLR + Introspection | 73.32% |
| Augment Noise + Introspection | 77.98% |
| Augmix + Introspection | 89.89% (ECE 43.33% ↓) |

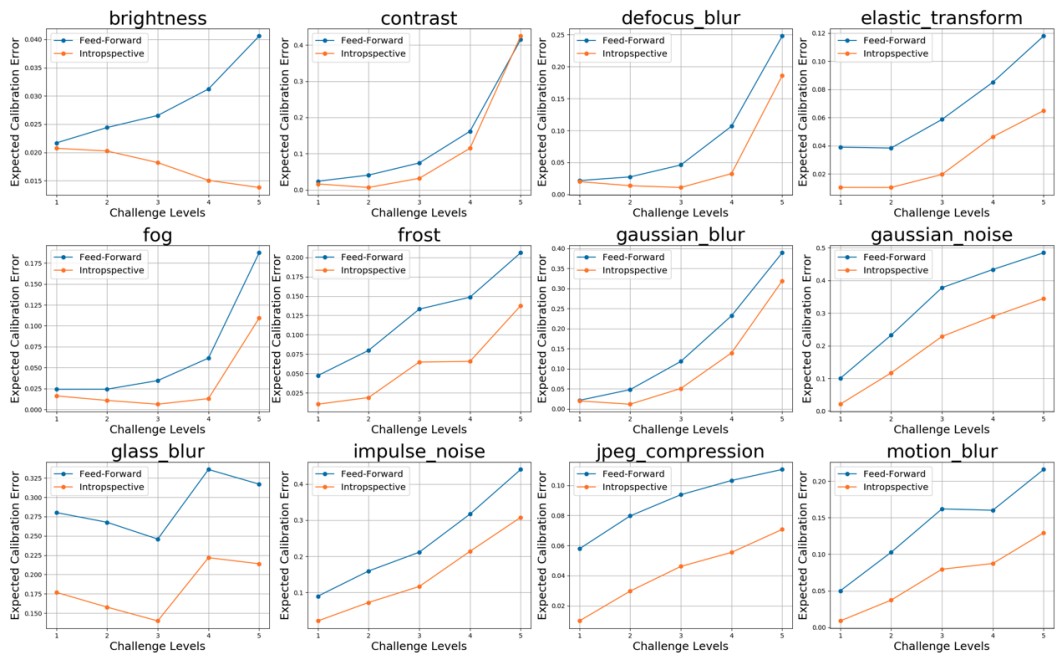

Figure 10: ECE vs distortion levels across 12 separate distortions from CIFAR-10C for ResNet-18.

### C.2.3 Introspection as a plug-in on top of existing techniques

Several techniques exist that boost the robustness of neural networks to distortions. These include training with noisy images (28), training with adversarial images (27), and self-supervised methods like SimCLR (19) that train by augmenting distortions. Another commonly used technique is to pre-process the noisy images to denoise them. All these techniques can be used to train $f(\cdot)$. Our proposed framework sits on top of any $f(\cdot)$. Hence, it can be used as a plug-in network. These results are shown in Table 5. Denoising 19 distortions is not a viable strategy assuming that the characteristics of the distortions are unknown. We use Non-Local Means denoising and the results obtained are lower than the feed-forward accuracy by almost $3\%$. However, introspecting on this model increases the results by $3.84\%$. We create untargeted adversarial images using I-FGSM attack with $\alpha = 0.01$ and use them to train a ResNet-18 architecture. In our experiments this did not increase the feed-forward accuracy. Introspecting on this network provides a gain of $2.84\%$. SimCLR (19) and introspection on SimCLR is discussed in Section C.3. In the final experimental setup of augmenting noise (28), we augment the training data of CIFAR-10 with six distortions - gaussian blur, salt and pepper, gaussian noise, overexposure, motion blur, and underexposure - to train a ResNet-18 network $f'(\cdot)$. We use the noise characteristics provided by (40) to randomly distort 500 CIFAR-10 training images by each of the six distortions. The original training set is augmented with the noisy data and trained. The results of the feed-forward $f'(\cdot)$ show a substantial increase in performance to $76.86\%$. This is about $9\%$ increase from the original architecture. We show that introspecting on $f'(\cdot)$ provides a further gain in accuracy of $1.12\%$. Note that to train $\mathcal{H}(\cdot)$, we do not use the augmented data. We only use the original CIFAR-10 undistorted training set. The gain obtained is by introspecting on only the undistorted data, even though $f'(\cdot)$ contains knowledge of the distorted data. Hence, introspection is a plug-in approach that works on top of any network $f(\cdot)$ or enhanced network $f'(\cdot)$. Augmix (26) is currently the best performing technique on CIFAR-10C. It creates multiple chains of augmentations to train the base WideResNet network. On CIFAR-10C, $f'(\cdot)$ obtains $89.85\%$ recognition accuracy. We use $f'(\cdot)$ as our base sensing model and train an introspective MLP on $f'(\cdot)$. Note that we do not use any augmentations for training $\mathcal{H}(\cdot)$. Doing so, we obtain a statistically similar accuracy performance of $89.89\%$. However, the expected calibration error of the feed-forward $f'(\cdot)$ model decreases by $43.33\%$ after introspection. Hence, when there is no accuracy gains to be had, introspection provides calibrated models.

Table 6: Expected Calibration Error and Maximum Calibrated Error for Feed-Forward vs Introspective Networks.

| Architectures | | ResNet-18 | ResNet-34 | ResNet-50 | ResNet-101 |
|---|---|---|---|---|---|
| ECE ($\downarrow$) | $f(\cdot)$ | 0.14 | 0.18 | 0.13 | 0.16 |
| | $\mathcal{H}(\cdot)$ | **0.07** | **0.09** | **0.06** | **0.1** |
| MCE ($\downarrow$) | $f(\cdot)$ | 0.27 | 0.34 | 0.27 | 0.32 |
| | $\mathcal{H}(\cdot)$ | **0.23** | **0.24** | **0.25** | **0.23** |

#### C.2.4 Expected Calibration Error (ECE)

In Fig. 5b), we show ECE for two distortion types - brightness and saturation across 5 distortion levels. In Fig. 10, we show results across five distortion levels for the first 12 distortions. The blue plot is the Feed-Forward ECE while the lower orange plot is its introspective counterpart. Apart from Level 5 contrast, intrsopective ResNet-18 is more calibrated than its feed-forward counterpart. This is in addition to the performance gains. The trend remains the same in the remaining distortions and among all considered networks. We average out ECE across 19 distortions and 5 challenge levels and provide ECE results for ResNets-18, 34, 50, 101 in Table 6. Lower the error, better is the architecture. The proposed introspective framework decreases the ECE of its feed-forward backbone by approximately $42\%$. An additional metric called Maximum Calibration Error (MCE) is also used for comparison. While ECE averages out the calibration difference in all bins (From Section 6), MCE takes the maximum error among all bins (37). The introspective networks outperform their feed-forward backbones among all architectures when compared using ECE and MCE.

Table 7: SimCLR and its supervised and introspective variations tested on CIFAR-10C.

| Methods | ResNet-18 | ResNet-34 | ResNet-50 | ResNet-101 |
|---|---|---|---|---|
| SimCLR (19) | 70.28% | 69.5% | 67.32% | 64.68% |
| SimCLR-MLP | 72.79% | 72.54% | 70.37% | 70.89% |
| SimCLR-Introspective (Proposed) | **73.32%** | **73.06%** | **71.28%** | **71.76%** |

#### C.3 SimCLR and Introspection

SimCLR (19) is a self-supervised contrastive learning framework that is robust to noise distortions. The algorithm involves creating augmentations of existing data including blur, noise, rotations, and jitters. The network is made to contrast between all the augmentations of the image and other images in the batch. A separate network head $g(\cdot)$ is placed on top of the network to extract features and inference is made by creating a similarity matrix to a feature bank. Note that $g(\cdot)$ is a simple MLP. Our proposed framework is similar to SimCLR in that we extract features and use an MLP $\mathcal{H}(\cdot)$ to infer from these features. In Table 5, we show the results of Introspecting ResNets against SimCLR. However, this comparison is unfair since the features in SimCLR are trained in a self-supervised fashion. In this section, we train SimCLR for ResNets-18, 34, 50, 101 and train a new MLP $g(\cdot)$, not for extracting features, but to classify images. In other words, in (19), the authors create $g(\cdot)$ to be a $512 \times 128$ layer that extracts features. We train a network of the form $512 \times 128 - 128 \times 10$ that is trained to classify images. We then introspect on this $g(\cdot)$ to obtain $r_x$. Hence, our extracted features are a result of introspecting on self-supervision. Note that $g(\cdot)$ is now a fully supervised network. We pass CIFAR-10C through $g(\cdot)$ and name it SimCLR-MLP in Table 7. It is unsurprising that the fully-supervised SimCLR-MLP beats the self-supervised SimCLR across all four ResNets. The introspective network is called SimCLR-Introspective in Table 7. Note that there is less than $1\%$ recognition performance increase across networks compared to SimCLR-MLP. Hence, the performance gains for introspecting on SimCLR-MLP is not as high as base ResNet architectures from Table 5. One hypothesis for this marginal increase is that the notions created within SimCLR-MLP are predominantly from the self-supervised features in SimCLR. These may not be amenable for the current framework of introspection that learns to contrast between classes and not between features within-classes.

#### C.4 Ablation Studies

The feature generation process in Section 2 is dependent on the loss function $J(\hat{y}, y)$. In this section, we analyze the performance of our framework for commonly used loss functions and show that the

Table 8: Introspective Learning accuracies when $r_x$ is extracted with different loss functions for ResNet-18 on CIFAR-10C.

| Feed-Forward | MSE-M | CE | BCE | L1 | L1-M | Smooth L1 | Smooth L1-M | NLL | SoftMargin |
|---|---|---|---|---|---|---|---|---|---|
| 67.89% | **71.4%** | 69.47% | 70.76% | 70.12% | 70.72% | 70.42% | 70.63% | 70.93% | 70.91% |

Table 9: Ablation studies for $\mathcal{H}(\cdot)$ on CIFAR-10C.

| | **Part 1 : Varying the number of layers** | |
|---|---|---|
| | Feed-Forward $64 \times 10$ | 67.89% |
| | $640 \times 10$ | 71% |
| R-18 | $640 \times 100 - 100 \times 10$ | **71.57%** |
| | $640 \times 300 - 300 \times 100 - 100 \times 10$ | 71.4% |
| | $640 \times 400 - 400 \times 200 - 200 \times 100 - 100 \times 10$ | 66.1% |
| | Feed-Forward $64 \times 10$ | 71.8% |
| R-50 | $2560 \times 300 - 300 \times 100 - 100 \times 10$ | **75.2%** |
| | $2560 \times 1000 - 1000 \times 500 - 500 \times 300 - 300 \times 100 - 100 \times 10$ | 73% |
| | **Part 2 : Is the performance increase only because of a large $\mathcal{H}(\cdot)$?** | |
| | Feed-Forward | 67.89% |
| | $f_L(\cdot)$ 1 Layer : $64 \times 10$ | 67.86% |
| | $\mathcal{H}(\cdot)$ 1 Layer : $640 \times 10$ | **71%** |
| | $f_L(\cdot)$ 3 Layers $64 \times 30 - 30 \times 20 - 20 \times 10$ | 63.61% |
| | $f_L(\cdot)$ 3 Layers $64 \times 512 - 512 \times 256 - 256 \times 10$ | 64.78% |
| | $\mathcal{H}(\cdot)$ 3 Layers: $640 \times 300 - 300 \times 100 - 100 \times 10$ | **71.4%** |
| R-18 | $f_L(\cdot)$, 6200 parameters : $64 \times 50 - 50 \times 40 - 40 \times 20 - 20 \times 10$ | 66.85% |
| | $\mathcal{H}(\cdot)$, 6400 parameters : $640 \times 10$ | **71%** |
| | Prediction on $y_{feat}$ using 10-NN (No $f_L(\cdot)$) | 66.31% |
| | Prediction on $r_x$ using 10-NN (No $\mathcal{H}(\cdot)$) | **68.76%** |
| | **Part 3 : VGG-16** | |
| | Feed-Forward | 68.96% |
| VGG-16 | $f(\cdot)$ $512 \times 1024 - 1024 \times 256 - 256 \times 10$ | 62.43% |
| | $\mathcal{H}(\cdot)$ $5120 \times 1000 - 1000 \times 100 - 100 \times 10$ | **73.79%** |
| | **Part 4 : Effect of approximation of Lemma 1 and Theorem 1** | |
| | Feed-Forward | $67.89 \pm 0.23\%$ |
| R-18 | With Approximation | $71.57 \pm 0.12\%$ |
| | Without Approximation | $71.43 \pm 0.11\%$ |

introspective network outperforms its feed-forward counterpart under any choice of $J(\hat{y}, y)$. We also ascertain the effect of the size of the parameter set in $\mathcal{H}(\cdot)$ on performance accuracy.

### C.4.1 Effect of Loss functions

We extract $r_x$ using 9 loss functions and report the final distortion-wise level-wise averaged results Table 8. We do so for ResNet-18 and for the architecture of $\mathcal{H}(\cdot)$ shown in Table 4. The following loss functions are compared : CE is Cross Entropy, MSE is Mean Squared Error, L1 is Manhattan distance, Smooth L1 is the leaky extension of Manhattan distance, BCE is Binary Cross Entropy, and NLL is Negative Log Likelihood. Notice that the performance of $r_x$ extracted using all loss functions exceed that of the feed-forward performance. The shown results of MSE, L1-M and Smooth L1-M are obtained by backpropagating a $\mathbf{1}_N$ from Theorem 1 vector multiplied by the average of all maximum logits $M$, in the training dataset. We use $M$ instead of 1 because we want the network to be as confidant of the introspective label $y_I$ as it is with the prediction label $\hat{y}$. Note that the results in Table 8 are for CIFAR-10C. MSE-M outperforms NLL loss by $0.37\%$ in average accuracy and is used in our experiments.

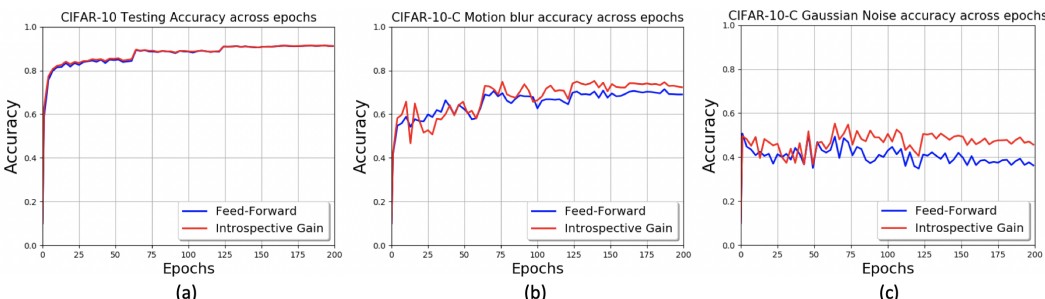

Figure 11: Introspective vs. Feed-Forward accuracy of ResNet-18 across training epochs on (a) CIFAR-10 original testset, (b) CIFAR-10C Motion Blur Testset on all 5 challenge levels, (c) CIFAR-10C Gaussian Noise Testset on all 5 challenge levels.

### C.4.2 Effect of $\mathcal{H}(\cdot)$

We conduct ablation studies to empirically show the following : 1) the design of $\mathcal{H}(\cdot)$ does not significantly vary the introspective results, 2) the extra parameters in $\mathcal{H}(\cdot)$ are not the cause of increased performance accuracy.

**How does changing the structure of $\mathcal{H}(\cdot)$ change the performance?** We vary the architecture of $\mathcal{H}(\cdot)$ from a single linear layer to 4 layers in the first half of Table 9 for ResNet-18. The results in the first three cases are similar. A four layered network performs worse than $f(\cdot)$. However, changing the weight decay from $5e^{-3}$ to $5e^{-4}$ during training increases the results to above $70\%$ but does not beat the smaller networks. For ResNet-18 architecture, the highest results are obtained when $\mathcal{H}(\cdot)$ is a 2-layered architecture but for the sake of uniformity, we use the results from a 3-layered network across all ResNet architectures.

**Are the extra parameters in $\mathcal{H}(\cdot)$ the only cause for increase in performance accuracy?** We show an ablations study of the effect of structure of $\mathcal{H}(\cdot)$ and $f(\cdot)$ on the introspective and feed-forward results in Part 2 of of Table 4 on CIFAR-10-C dataset. The results are divided into four sections. In the first section, we show the performance of the original feed-forward network $f(\cdot)$, the performance when the final layer, $f_L(\cdot)$ is retrained using features $y_{feat}$ from Eq. 1, and the introspective network when $\mathcal{H}(\cdot)$ is a single layer. The second section shows the results when the features $y_{feat}$ are used to train a three layered network $f_L(\cdot)$, and the introspective network is also three layered. Finally, in section 3, we try to equate the number of parameters for $f_L(\cdot)$ and $\mathcal{H}(\cdot)$. Note that in all cases, $f_L(\cdot)$ and $\mathcal{H}(\cdot)$ are trained in the same manner as detailed in Section C.1. $\mathcal{H}(\cdot)$ beats the performance of $f_L(\cdot)$ among all ablation studies. Finally, similar to SimCLR, we forego using an MLP and use 10-Nearest Neighbors on $y_{feat}$ ($64 \times 1$) and $r_x$ ($640 \times 1$) for predictions. Both results are worse-off than their MLP results but $r_x$ outperforms $y_{feat}$.

**Introspection on larger $r_x$ and wider $f_{L-1}(\cdot)$** On Resnet-101 experiments in Section C.2.1, $r_x$ is of dimension $2560 \times 1$. On all Resnet-18 experiments, $r_x$ is $640 \times 1$. In part 3 of Table 4, we show results on a larger VGG-16 architecture where $r_x$ is of size $5120 \times 1$. Row 1 shows the normal feed-forward accuracy. Row 3 is the introspective results on VGG-16 and the results are $4.83\%$ higher than its feed-forward counterpart. In row 2, we expand the penultimate layer of $f(\cdot)$ from 512 to 1024 so as to include more parameters in the feed-forward network. Note that this is not possible for Resnet-18 as $f_{L-1}(\cdot)$ is $64 \times 1$. Hence, we compare our introspective network against a wider $f(\cdot)$ architecture instead of an elongated architecture like in Part 2. In both cases, introspection outperforms additional parameters in $f(\cdot)$.

**Effect of Lemma 1 and Theorem 1** Note that with approximations from Lemma 1 and Theorem 1, $r_x$ generation occurs with a time complexity of $\mathcal{O}(1)$. Without approximation, $r_x$ generation occurs in $\mathcal{O}(N)$. However, the results are statistically insignificant when averaged across 5 seeds.

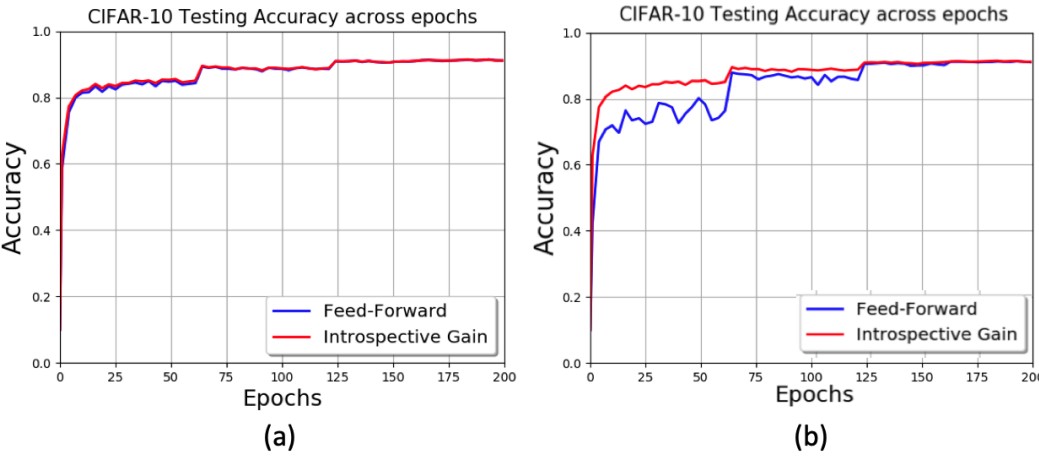

Figure 12: Introspective vs. Feed-Forward accuracy of ResNet-18 across training epochs when (a) $f(\cdot)$ and $\mathcal{H}(\cdot)$ are trained on the same training set (b) $\mathcal{H}(\cdot)$ is trained on a separate held-out validation set

### C.5   Introspective accuracy across training epochs

In Section 2, we make the assumption that $f(\cdot)$ is well trained to approximate $r_x$ using Theorem 1. In Section 4, the Fisher Vector analysis works when the gradients form distances across the manifold in $f(\cdot)$ which occurs if $f(\cdot)$ is well trained. In this section we show that, practically, introspection performs as well as feed-forward accuracy across training epochs on CIFAR-10 testset and outperforms feed-forward accuracy on CIFAR-10C distortions. We show results on original testset, gaussian noise and motion blur testsets in Fig. 11.

**Training, Testing, and Results in Fig. 11a**   In this experimental setup, ResNet-18 is trained for 200 epochs. The model states at multiples of 3 epochs from 1 to 200 are stored. This provides 67 states of $f(\cdot)$ along its training process. Each $f(\cdot)$ is tested on CIFAR-10 testset and the recognition accuracy is plotted in blue in Fig. 11a). The introspective features $r_x$ for all 67 states are extracted for the $50,000$ training samples. These $r_x$ are used to train 67 separate $\mathcal{H}(\cdot)$ of structure provided in Table 4 with a similar training setup as in Section C.1. The $r_x$ from the $10,000$ testing samples are extracted individually for each of the 67 $f(\cdot)$ states and tested. The results are plotted in red in Fig. 11a). Note the sharp spikes at epochs 60 and 120 where there is a change in the learning rate. Hence, when training and testing distributions are similar, introspective and feed-forward learning provides statistically similar performance across varying states of $f(\cdot)$.

**Training, Testing, and Results in Fig. 11b, c**   We now consider the case when a network $f(\cdot)$ is trained on distribution $\mathcal{X}$ and tested on $\mathcal{X}'$ from CIFAR-10C distortions. The 67 trained models of ResNet-18 are tested on two distortions from CIFAR-10C. From the results in Fig. 5, introspective learning achieves one of its highest performance gains in Gaussian noise, and an average increase in motion blur after epoch 200. The results in Fig. 11 indicate that after approximately 60 epochs, the feed-forward network has sufficiently sensed notions to reflect between classes. This is seen in the performance gains in both the motion blur and Gaussian noise experiments.

**Training of $\mathcal{H}$ on a separate validation set in Fig. 12b**   In all experiments, the introspective network $\mathcal{H}(\cdot)$ is trained on the same training set as $f(\cdot)$. In Fig. 12, we show the results when the introspective network is trained on a separate portion of the dataset. We use $40,000$ images to train $f(\cdot)$ and $10,000$ to train $\mathcal{H}(\cdot)$ both of which are randomly chosen. We follow the training procedure from before. The model states at multiples of 3 epochs from 1 to 200 are stored. This provides 67 states of $f(\cdot)$ along its training process. Each $f(\cdot)$ is tested on CIFAR-10 testset and the recognition accuracy is plotted in blue in Fig. 12b). The $\mathcal{H}(\cdot)$ at each iteration on the other hand is trained with the $10,000$ images. However, it has access to the notions created from the remaining $40,000$ images and hence the results for introspection match Fig. 11a) which is reproduced in Fig. 12a). The feed-forward results catch up to the introspective results around epoch 60. At Epoch 120, we add

Table 10: Performance of Proposed Introspective $\mathcal{H}(\cdot)$ vs Feed-Forward $f(\cdot)$ Learning under Domain Shift on Office dataset

| Architectures | | DSLR ↓ Amazon | DSLR ↓ Webcam | Amazon ↓ DSLR | Amazon ↓ Webcam | Webcam ↓ DSLR | Webcam ↓ Amazon |
|---|---|---|---|---|---|---|---|
| ResNet-18 | $f(\cdot)$ | 39.1 | 78 | 62.9 | 59 | 89.8 | 42.2 |
| (%) | $\mathcal{H}(\cdot)$ | **47** | **90.7** | **67.3** | **63.9** | **96** | **44** |
| ResNet-34 | $f(\cdot)$ | 41.8 | 83.3 | **67.3** | 60.1 | 90.6 | 41.7 |
| (%) | $\mathcal{H}(\cdot)$ | **46.4** | **89.8** | **67.3** | **63.9** | **97.8** | **43.3** |
| ResNet-50 | $f(\cdot)$ | - | - | 67.3 | 62 | 92.4 | 33.4 |
| (%) | $\mathcal{H}(\cdot)$ | - | - | **78.1** | **68.4** | **97.8** | 30.8 |
| ResNet-101 | $f(\cdot)$ | - | - | 62.9 | 59 | 89.8 | 31.77 |
| (%) | $\mathcal{H}(\cdot)$ | - | - | **76.5** | **67.3** | **92.4** | **33.6** |

back the $10,000$ held-out images into the training set of $f(\cdot)$ and the results match between Fig. 12a) and Fig. 12b).

Table 11: Performance of Proposed Introspective $\mathcal{H}(\cdot)$ vs Feed-Forward $f(\cdot)$ Learning under Domain Shift on VisDA Dataset

| ResNet-18 | Plane | Cycle | Bus | Car | Horse | Knife | Bike | Person | Plant | Skate | Train | Truck | All |
|---|---|---|---|---|---|---|---|---|---|---|---|---|---|
| $f(\cdot)$ (%) | 27.6 | 7.2 | **38.1** | 54.8 | 43.3 | **4.2** | **72.7** | **8.3** | 28.7 | 22.5 | **87.2** | 2.9 | 38.1 |
| $\mathcal{H}(\cdot)$ (%) | **39.9** | **27.6** | 19.6 | **79.9** | **73.5** | 2.7 | 46.6 | 6.5 | **43.8** | **30** | 73.6 | **4.3** | **43.58** |

## C.6 Results on large images

### C.6.1 Domain Adaptation on Office dataset

In Section 4, we claim that introspection helps a network to better classify distributions that it has not seen while training. In Section 6, we tested on 95 new distributions in CIFAR-10C and 30 new distributions in CIFAR-10-CURE. In this section, we evaluate the efficacy of introspection when there is a domian shift between training and testing data under changes in background, and camera acquisition setup among others. Specifically, the robust recognition performance of $\mathcal{H}(\cdot)$ is validated on Office (42) dataset using Top-1 accuracy. The Office dataset has 3 domains - images taken from either Webcam or DSLR, and extracted from Amazon website. Images can belong to any of 31 classes and they are of varying sizes - upto $1920 \times 1080$. Hence, results on Office shows the applicability of introspection on large resolution images. ImageNet pre-trained ResNet-18,34,50,101 (4) architectures are used for $f(\cdot)$. The final layer is retrained using the source domain while the remaining two domains are for testing. The experimental setup, the same detailed in Section 6, is applied and the Top-1 accuracy is calculated. The results are summarized in Table 10. In every instance, the top domain is $\mathcal{X}$ - the training distribution, and the bottom domain is $\mathcal{X}'$ - the testing distribution. Note that ResNet-50 and 101 failed to train on 498 images in DSLR source domain. The results of introspection exceed that of feed-forward learning in all but ResNet-50 when classifying between Webcam and Amazon domains.

### C.6.2 Domain Adaptation on Vis-DA dataset

Validation results on a synthetic-to-real domain shift dataset called VisDA (59) are presented in Table 11. VisDA has 12 classes with about $152,000$ synthetic training images, and $55,000$ real validation images. The validation images are cropped images from MS-COCO. ResNet-18 architecture pretrained on ImageNet is finetuned on the synthetically generated training images from VisDA dataset fro 200 epochs. It is then tested on the validation images and the recognition performance is shown in Table 11 as feed-forward $f(\cdot)$ results. Introspective $\mathcal{H}(\cdot)$ results are obtained and shown when $f(\cdot)$ is ResNet-18. There is an overall improvement of $5.48\%$ in terms of performance accuracy. However, the individual class accuracies leave room for improvement.

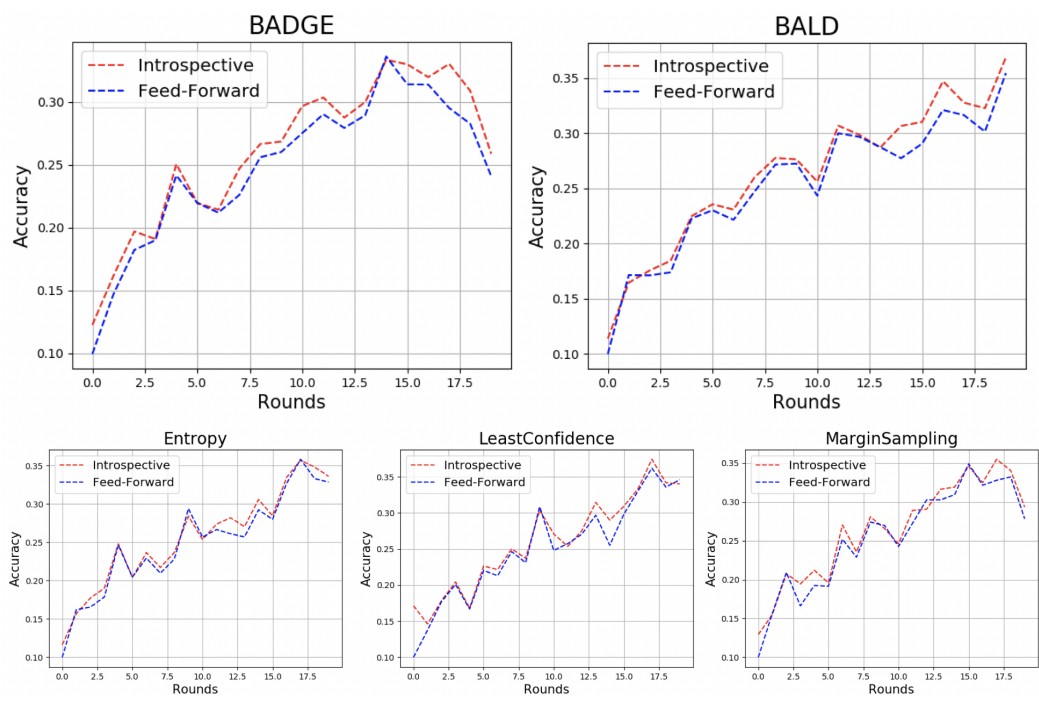

Figure 13: Introspective vs. Feed-Forward accuracy of ResNet-18 across training rounds for state-of-the-art techniques in an active learning setting. The query batch size per round is 1000. The trainset is CIFAR-10 and testset is Gaussian Noise from CIFAR-10C.

Table 12: Out-of-distribution Detection of existing techniques compared between feed-forward and introspective networks when the data is under adversarial attack.

| Methods | OOD Datasets (Attack) | FPR (95% at TPR) ↓ | Detection Error ↓ | AUROC ↑ |
|---|---|---|---|---|
| | | Feed-Forward/Introspective | | |
| MSP [35] | Textures | 99.98/**23.19** | 45.9/**7.9** | 30.4/**96.48** |
| | iSUN | 98.63/**87.2** | 46.71/**28.95** | 46.44/**75.81** |
| | Places365 | 100/**83.59** | 47.64/**26.46** | 25.08/**79** |
| | LSUN-C | 99.65/**87.64** | 43.38/**26.31** | 43.47/**78.4** |
| ODIN [36] | Textures | 99.95/**2.06** | 47.7/**3.48** | 37.5/**99.11** |
| | iSUN | 96.8/**90.42** | 44.77/**31.11** | 53.88/**73.22** |
| | Places-365 | 99.97/**82.5** | 47.12/**26.86** | 32.69/**78.88** |
| | LSUN-C | 98.6/**88.28** | 40.51/**27.88** | 56.7/ **77.25** |

# D   Appendix: Downstream Applications

## D.1   Active Learning

In Table 2, the mean recognition accuracies across the first 20 rounds of Active Learning experiments for commonly used query strategies are shown. We plot these recognition accuracies across for all five query strategies in Fig. 13. The x-axis is the round at which the performance is calculated. The calculated accuracy is plotted on the y-axis. The experiment starts with a random 100 images in round 1. Each strategy queries using either a round-wise sample trained $f(\cdot)$ or a round-wise sample trained $\mathcal{H}(\cdot)$. Note that at each round, the networks are retrained. This continues for 20 rounds. Both BALD [34] and BADGE [33] applied on $\mathcal{H}(\cdot)$ consistently beat its $f(\cdot)$ counterpart on every round. This is because both these methods rely on extracting features from the network as compared to the other three techniques that directly use the output logits from either $\mathcal{H}(\cdot)$ or $f(\cdot)$. Since the network is not well-trained at the initial stages - due to a dearth of training data - the introspective network is not as consistent as the feed-forward network among Entropy, Least Confidence, and Margin strategies. Nonetheless, $\mathcal{H}(\cdot)$ outperforms $f(\cdot)$ on average across all rounds.

Table 13: Performance of Contrastive Features against Feed-Forward Features and other Image Quality Estimators. Top 2 results in each row are highlighted.

| Database | PSNR HA | IW SSIM | SR SIM | FSIMc | Per SIM | CSV | SUM MER | Feed-Forward UNIQUE | Introspective UNIQUE |
|---|---|---|---|---|---|---|---|---|---|
| | | | | | Outlier Ratio (OR, ↓) | | | | |
| **MULTI** | 0.013 | 0.013 | **0.000** | 0.016 | 0.004 | **0.000** | **0.000** | **0.000** | **0.000** |
| **TID13** | **0.615** | 0.701 | 0.632 | 0.728 | 0.655 | 0.687 | **0.620** | 0.640 | **0.620** |
| | | | | | Root Mean Square Error (RMSE, ↓) | | | | |
| **MULTI** | 11.320 | 10.049 | 8.686 | 10.794 | 9.898 | 9.895 | **8.212** | 9.258 | **7.943** |
| **TID13** | 0.652 | 0.688 | 0.619 | 0.687 | 0.643 | 0.647 | 0.630 | **0.615** | **0.596** |
| | | | | | Pearson Linear Correlation Coefficient (PLCC, ↑) | | | | |
| **MULTI** | 0.801 | 0.847 | 0.888 | 0.821 | 0.852 | 0.852 | **0.901** | 0.872 | **0.908** |
| | -1 | -1 | 0 | -1 | -1 | -1 | -1 | -1 | |
| **TID13** | 0.851 | 0.832 | 0.866 | 0.832 | 0.855 | 0.853 | 0.861 | **0.869** | **0.877** |
| | -1 | -1 | 0 | -1 | -1 | -1 | 0 | -1 | |
| | | | | | Spearman's Rank Correlation Coefficient (SRCC, ↑) | | | | |
| **MULTI** | 0.715 | **0.884** | 0.867 | 0.867 | 0.818 | 0.849 | **0.884** | 0.867 | **0.887** |
| | -1 | 0 | 0 | 0 | -1 | -1 | 0 | 0 | |
| **TID13** | 0.847 | 0.778 | 0.807 | 0.851 | 0.854 | 0.846 | 0.856 | **0.860** | **0.865** |
| | -1 | -1 | -1 | -1 | 0 | -1 | -1 | | |
| | | | | | Kendall's Rank Correlation Coefficient (KRCC) | | | | |
| **MULTI** | 0.532 | **0.702** | 0.678 | 0.677 | 0.624 | 0.655 | 0.698 | 0.679 | **0.702** |
| | -1 | 0 | 0 | 0 | -1 | 0 | 0 | 0 | |
| **TID13** | 0.666 | 0.598 | 0.641 | 0.667 | **0.678** | 0.654 | 0.667 | 0.667 | **0.677** |
| | 0 | -1 | -1 | 0 | 0 | 0 | 0 | 0 | |

## D.2 OOD

**Adversarial setting in Table 3** A datapoint $z$, is perturbed as $z + \epsilon$ and the goal of the detector is to classify $z \in \mathcal{X}$ or $z \in \mathcal{X}'$. This modality is proposed by the authors in (44) and we use their setup. PGD attack with perturbation $0.0014$ is used. The same MSP and ODIN detectors from Table 3 are utilized. On 4 OOD datasets, both MSP and ODIN show a performance gain across all three metrics on $\mathcal{H}(\cdot)$ compared to $f(\cdot)$. Note that the results in Table 12 is for ResNet-18 architecture for the same $f(\cdot)$ and $\mathcal{H}(\cdot)$ used in other experiments including Fig. 4.

**Vanilla setting in Table 12** In Table 12, we show the results of out-of-distribution detection when $\mathcal{X}$ is CIFAR-10 and $\mathcal{X}'$ are the four considered datasets. Note that among the four datasets, textures and SVHN are more out-of-distribution from CIFAR-10 than the natural image datasets of Places365 and LSUN. The results of the introspective network is highest on Textures DTD dataset.

## D.3 Image Quality Assessment

**Related Works** Multiple methods have been proposed to predict the subjective quality of images including PSNR-HA (60), IW-SSIM (61), SR-SIM (62), FSIMc (63), PERSIM (64), CSV (65), SUMMER (66), ULF (67), UNIQUE (39), and MS-UNIQUE (68). All these methods extract structure related hand-crafted features from both reference and distorted images and compare them to predict the quality. Recently, machine learning models have been proposed to directly extract features from images (39). (39) propose UNIQUE that uses a sparse autoencoder trained on ImageNet to extract features from both reference and distorted images. We use UNIQUE as our base network $f(\cdot)$.

**Feed-Forward UNIQUE** (39) train a sparse autoencoder with a one layer encoder and decoder and a sigmoid non-linearity on $100,000$ patches of size $8 \times 8 \times 3$ extracted from ImageNet testset. The autoencoder is trained with MSE reconstruction loss. This network is our $f(\cdot)$. UNIQUE follows a full reference IQA workflow which assumes access to both reference and distorted images while estimating quality. The reference and distorted images are converted to YGCr color space and converted to $8 \times 8 \times 3$ patches. These patches are mean subtracted and ZCA whitened before being passed through the trained encoder. The activations of all reference patches in the latent space are extracted and concatenated. Activations lesser than a threshold of $0.025$ are suppressed to $0$. The choice of threshold $0.025$ is made based on the sparsity coefficient used during training. Similar procedure is followed for distorted image patches. The suppressed and concatenated features of both the reference and distorted images are compared using Spearman correlation. The resultant is the feed-forward estimated quality of the distorted image.

**Introspective-UNIQUE** We use the architecture and the workflow from (39) which is based on feed-forward learning to demonstrate the value of introspection. We replace the feed-forward features with the proposed introspective features. The loss in Eq. 6 for introspection is not between classes but between the image $x$ and its reconstruction $\tilde{x}$ from the sparse autoencoder from (39). For a reference image $x$, $r_x$ is derived using $J(x, \tilde{x})$. Hence, gradients of $r_x$ span the space of reconstruction noise. Since the need in IQA is to characterize distortions, we obtain $r_x$ for reference images from the first layer and project both reference and distorted images onto $r_x$. These projections are compared using Spearman correlation to assign a quality estimate. In this setting, $\mathcal{H}(\cdot)$ is the projection operator and Spearman correlation. Hence, Introspective-UNIQUE broadens introspection in the following ways - 1) defining introspection on generative models, 2) using gradients in the earlier layers of a network.

**Statistical Significance** We use the statistical significance code and experimental modality from the Feed-Forward model (39). Specifically, we follow the procedure presented for IQA statistical significance test regulations suggested by ITU-T Rec. P.1401. Normality tests are conducted on the human opinion scores within the datasets and those scores that significantly deviate from dataset-specific parameters are discarded. Hence, for the purpose of our statistical significance tests, we assume that the given scores are a good fit for normal distribution. The predicted correlation coefficients from the proposed Introspective-UNIQUE technique are compared individually against all other techniques in Table 13. For the test itself, we use Fisher-Z transformation to obtain the normally distributed statistic between the compared methods. A 0 corresponds to statistically similar performance between Introspective-UNIQUE and the compared method, ´-1 means that the compared method is statistically inferior to Introspective-UNIQUE, and 1 indicates that the compared method is statistically superior to Introspective-UNIQUE.

**Results** We report the results of the proposed introspective model in comparison with commonly cited methods Table 13. We utilize MULTI-LIVE (MULTI) (69) and TID2013 (70) datasets for evaluation. The performance is validated using outlier ratio (consistency), root mean square error (accuracy), Pearson correlation (linearity), Spearman correlation (rank), and Kendall correlation (rank). Arrows next to each metric in Table 13 indicate the desirability of a higher number (↑) or a lower number(↓). Two best performing methods for each metric are highlighted. The proposed framework is always in the top two methods for both datasets in all evaluation metrics. In particular, it achieves the best performance for all the categories except in OR and KRCC in TID2013 dataset. The feed-forward model does not achieve the best performance for any of the metrics in MULTI dataset. However, the same network using introspective features significantly improves the performance and achieves the best performance on all metrics. For instance, the feed-forward model is the third best performing method in MULTI dataset in terms of RMSE, PLCC, SRCC, and KRCC. However, the introspective features improve the performance for those metrics by 1.315, 0.036, 0.020, and 0.023, respectively and achieve the best performance for all metrics. This further reinforces the plug-in capability of the proposed introspective inference. Additionally, against no technique is our method not statistically significant in at least 1 metric.

### D.4 Uncertainty

Table 14: Uncertainty quantification algorithms measured against Introspective Resnet-18. Accuracy is recognition accuracy.

| Modality | Techniques | CIFAR-10-Rotations | | |
|---|---|---|---|---|
| | | Log-likelihood (↑) | Brier Score (↓) | Accuracy (↑) |
| Ensemble | Bootstrap (71) | −2.03 | 0.71 | 0.47 |
| Bayesian | MC Dropout (72) | −2.81 | 0.85 | 0.44 |
| Bayesian | BBP without lrt (73) | −1.87 | 0.74 | 0.39 |
| Bayesian | BBP with lrt (73) | −2.33 | 0.78 | 0.42 |
| Deterministic | TENT (74) | −15.54 | 1.36 | 0.30 |
| Deterministic | Introspective Resnet-18 | −2.97 | 0.89 | 0.45 |

Existing methods of uncertainty quantification are compared against an Introspective Resnet-18 in Table. 14. Column 1 denotes the modality of the method. A deterministic method like the proposed introspection and TENT (74) require only a single pass through the network. Bayesian

networks require multiple passes. Ensemble techniques require multiple networks. We use two uncertainty measures, log-likelihood and brier score, to measure uncertainty. A detailed review of these measures is presented in (71). For all methods, we use the same base Resnet-18 trained on the undistorted images from Section 6 as our architecture. A rotated version of CIFAR-10 testset is used to determine uncertainty. For every image there are 16 rotated versions of the same image. Hence, there are 160, 000 images in the testset. The results are presented in Table 14. The ensemble method outperforms all other techniques both interms of accuracy as well as uncertainty. However, the proposed introspective technique performs comparably to bayesian techniques among the uncertainty metrics while beating them in recognition accuracy. It outperforms TENT among both recognition accuracy and uncertainty metrics.

# E   Appendix: Reproducibility Statement

The paper uses publicly available datasets to showcase the results. Our introspective learning framework is built on top of existing deep learning apparatus - including ResNet architectures (4) (inbuilt PyTorch architectures), CIFAR-10C data (27) (source code at `https://zenodo.org/record/2535967#.YLpTF-1KhhE`), calibration ECE and MCE metrics (37) (source code at `https://github.com/markus93/NN_calibration`), out-of-distribution detection metrics, and codes for existing methods were adapted from (44) (source code at `https://github.com/jfc43/robust-ood-detection`), active learning methods and their codes were adapted from (33) (source code at `https://github.com/JordanAsh/badge`), Grad-CAM was adapted from (6) (code used is at `https://github.com/adityac94/Grad_CAM_plus_plus`). Our own codes will be released upon acceptance. The exact training hyperparameters for $f(\cdot)$ and $\mathcal{H}(\cdot)$, and all considered $\mathcal{H}(\cdot)$ architectures are shown in Appendix C.1. Extensive ablation studies on $\mathcal{H}(\cdot)$ are shown in C.4.