# OpenReview forum: "Introspective Learning : A Two-Stage approach for Inference in Neural Networks"
_NeurIPS.cc/2022/Conference — NeurIPS 2022 Accept_

### Official Review · Reviewer_8XSC · 2022-07-08

**Rating:** 3
**Confidence:** 4
**Soundness:** 2 fair
**Presentation:** 3 good
**Contribution:** 2 fair

**Summary:**

The paper advocates the use of a two stage process for classification to increase robustness of NN. The first net is standard. The second network leverages gradients of the first network as inputs, i.e., they compute gradients for all possible outcomes towards the last layer (at least this is what seems to be the case according to Fig. 3).


**Questions:**

None

**Limitations:**

Limitations are ok.

**Strengths And Weaknesses:**

- The basic idea is not novel against the authors claim. See [1]. This work published in 2020 uses also a 2-stage process (one fast and one slow stage), it speaks of reflection, uses the Kahnemann reference, etc. However, aside from these similarities, the technical details seem to be quite different. Also the cited work does not rely on explanations of all classes. However, the authors should discuss this work.
- Theory is appreciated, but I have doubt on correctness. In the proof supplementary material from equation (15) to (16) the authors transform using log(a0*b0+a1*b1)= log(a0+a1) +log(b0+b1). But this equation is not true.
- The paper does not improve in general, i.e., for regular CIFAR-10, but only for CIFAR-10C. Thus, it seems valuable only in a few cases (e.g. the work above improves on CIFAR-10 on the test set using reflection)
- The evaluation is just on one dataset and one architecture. This is insufficient and makes the generalization very questionable.

Detailed comments:
- Robsutness  -> Robustness
[1] "Reflective-Net: Learning from Explanations" from 2020 (https://arxiv.org/abs/2011.13986)

---

> ### Author Response · Authors · 2022-08-03
> **Theory correctness; Motivation of paper**
>
> We thank the reviewer for their comments. We have addressed each comment below while also uploading a new manuscript that takes the comments into consideration. The changes are in blue
>
> **Reflective-Net** We were unaware of this work and thank the reviewer for bringing it to our attention. We have cited it in two places in the revised manuscript: 1) To motivate reflection in neural networks in Line 43, 2) As a reference for two-stage networks in line 88. As the reviewer notes, the technical details between the works differ in two ways:
>
> 1. The introspective questions that are being reflected on is different between the two works. Our technique relies on $N$ contrastive questions that are a function of both the network's prediction and the introspective class.
>
> 2. The features that are used as a measure of reflection. Our technique directly takes the change in model parameters (gradients) rather than activations that are guided by gradients (Grad-CAM). These produce sparse and robust features.
>
> **Theory correctness** We thank the reviewer for pointing out an error in the proof. Please note, from Eqs. (15)-(16), we do not use $log(a_0 a_1 + b_0 b_1) = log(a_0+a_1) + log(a_1 + b_1)$. Rather we use the product of logarithms, - $log (a(\sum_j b_j)) = log a + log (\sum_j{b_j})$. Note that in Eq. 15, $a = y_{\hat{y}}$ and is a constant. We take this out of the summation in Eq. 16 in the revised manuscript as an intermediate step. Then, we use the product of logarithms to separate $a$ from the summation in Eq. 17. In the previous version of the manuscript, we had erroneously included the summation along with $a$ that led to the confusion. There is no summation in the second term and we have addressed this. Note that we also empirically show this theory in the following ways:
>
> 1. In Fig. 2, we show results on the last layer of a network trained on MNIST. This is a *true* well-trained network in the sense that the predicted logit is high compared to all the other logits. The only changes are in the weights associated with the predicted logit $y_\hat{y}$ and the introspective logit $y_I$.
>
> 2. We show the same results in Fig. 7 on a network trained on CIFAR 10. This network is not as well trained as MNIST (from an approximation sense) and hence the visualization in Fig. 7 is not as sparse as Fig. 1.
>
> 3. In Part 4 of Table 9, in the ablation study, we show the negligible effect of all the other gradients. The results do not differ with and without the approximation since most of the gradients that are not associated with $y_I$ and $y_{\hat{y}}$ are zero. It only adds to the complexity of the MLP.
>
> **Limited functionality: The paper does not improve in general, i.e., for regular CIFAR-10, but only for CIFAR-10C** The proposed technique demonstrates robust performance under distributional shift under deployment. We believe that this is a major contribution of the paper rather than a weakness. We explicitly motivate the contribution of our introspective network through multiple means - reasoning, biological, mathematical, and evaluation.
>
> 1. *Reasoning:* We motivate abductive reasoning and introspective questions in Line 69 onwards. We expand on this further in the Appendix A, lines 614-623. In summary, since introspection deals with examining ones notion, it is best served by answering contrastive questions. And by definition, abductive reasoning helps humans generalize to new situations [54 in paper]. This motivates the paper to conduct multiple evaluations of experiments under distributional shift (which we define as *new situations* for a network).
>
> 2. *Biological:* Recognition is inherently a feed-forward process (it takes around 80ms). We only need to reflect on this feed-forward process when encountered with a surprising situation. We take this surprising situation to be a new one under distributional shift. We expand on this under Biological plausibility of Introspection in Lines 652 onwards. In summary, we posit that introspection is not required when there is no distributional shift and provide reasoning and cognitive science based reasons for this assertion.
>
> 2. *Mathematical:* Throughout the main paper, we have shown why introspective features are robust to distributional shifts. In Lines 159-164 we discuss the noise required in the input, to change the prediction. In Lines 201-217, we explicitly discuss the robustness of the $H(\cdot)$ network and state that $H(\cdot)$ does not affect the testset results (line 207). We theoretically establish its usage under distributional shift from lines 209-217.
>
> 4. *Evaluation:* Being robust to distributional shift does not decrease the merits of our framework. Infact, we provide a holistic evaluation of the distribution shift under four applications. We further discuss the evaluatory contributions of the paper in the next comment.

---

> > ### Author Response · Authors · 2022-08-03
> > **Evaluation scheme**
> >
> > *The evaluation is just on one dataset and one architecture. This is insufficient and makes the generalization very questionable.* Because of limited space, we had to push the bulk of our evaluation to the supplementary materials. Across Appendix C and D, we have multiple applications and ablation studies, including results on other datasets and networks.
> >
> > 1. In Appendix C.6, we show results for large-scale domain difference settings on Office and VisDA datasets. Office dataset has images of varying sizes including 1920x1080. Among both these datasets, introspection provides performance gains. Specifically on Office, we show results on ImageNet pre-trained ResNet-18, 34, 50, and 101 networks in Table 10. The resulting dimensionality of introspective features when trained on Amazon domain is more than 12 times the size of ResNet-18 trained on CIFAR-10. The number of classes to discriminate between are 31.
> >
> > 2. In Appendix D.1, we examine results in an active learning setting. All existing query strategies use the output logits to estimate the next batch of data to label. Hence, more calibrated and generalizable is the model, more effective is any query strategy. Akin to recognition experiments, all query strategies perform better under introspection on CIFAR-10C.
> >
> > 3. In Appendix D.2, we show results of Out-of-Distribution detection, when existing method like ODIN and baseline techniques are used on both feed-forward and introspective networks. We do so under two settings. The first is the normal OOD setting where the network is trained on CIFAR-10 and tested on one of four datasets – Textures, iSUN, Places365, and LSUN. The second setting is the harder adversarial setting where not only is the query data OD or ID, it is also under adversarial attack. In both these settings we see introspection providing performance gains.
> >
> > 4. In Appendix D.3, we examine human introspection on Image Quality Assessment (IQA). IQA requires humans to evaluate a noisy image based on its pristine version. Hence IQA most resembles our reflection setting. Datasets consist of noisy and pristine images, and quality scores. In IQA, we show that introspection provides statistically significant results on two datasets – MULTI-LIVE and TID2013.
> >
> > 5. In Appendix D.4, we show results on uncertainty estimation on rotated CIFAR-10R images. The best results are bayesian methods that use multiple models and outputs to create uncertainty measures. However, introspection (which is deterministic) provides comparable results to bayesian methods while performing better than TENT, another deterministic model.
> >
> > 6. We have multiple ablation studies in Table 9. In Figure 4, the results are presented on four networks and two datasets of varying noises. On a side note, AugMix introspection (Line 313) is conducted on WideResNet since we were unable to replicate good results of AugMix on ResNet-18. We used the authors code for the feed-forward network and introspected on top of it.
> >
> > *Evaluation strategy for a holistic view on distributional shift* We believe that our framework is better evaluated through the setup where we use the same dataset for training across related applications. An important contribution of the paper is that we induce networks to be more robust during deployment under new situations. We look at this task of deployment holistically by considering multiple applications – recognition, calibration, OOD, Active Learning, and Uncertainty. This helps us make claims like the following on Line 224 - The same framework that robustly recognizes images despite noise can also detect noise to make an out-of-distribution detection. In literature, CIFAR-10 is the common dataset among all these applications. While CIFAR-10C and CIFAR-10-CURE are useful for recognition, CIFAR-10R is used for uncertainty estimation. Hence, we analyze any trained network holistically as opposed to methods that contribute to one application. For instance, AugMix technique produces more robust accuracy on CIFAR-10C. However, it does not help with calibration. Introspection on top of AugMix not only conserves accuracy, but also reduces calibration error by 43% (Line 315). Hence, a holistic look at distributional shift requires all facets of distributional deployment issues to be addressed.
> >
> > This is not to say that larger datasets other than CIFAR-10 are immaterial. We also have large-scale images in Appendix C.6 and D.3 sections that show introspection on larger models and datasets. Appendix D.3 provides a completely new application that we believe most closely resembles introspection.
> >
> > We are happy to further discuss with the reviewer regarding the evaluation strategy. Since NeurIPS allows an additional page for the accepted version, we have added both the Active Learning and OOD experiments to the main paper. Please note that these are not new experiments but are taken from the supplementary materials that was submitted earlier.

---

> > > ### Comment · Reviewer_8XSC · 2022-08-08
> > > **...**
> > >
> > > Thanks for your reponse. It is good and also raises a few general issues like what should be the page limit.
> > >
> > > Key points should be in the paper. For me adding a few lines showing more datasets is among them. The discussion of them could be in an Appendix.
> > >
> > > Also I think one should not be very forgiving in terms of errors.
> > >
> > > Also other concerns like the lack of improvement in general remains. The authors have different opinion on the value, which is ackknowledged but does not change much.

---

### Official Review · Reviewer_W6Yu · 2022-07-11

**Rating:** 6
**Confidence:** 3
**Soundness:** 3 good
**Presentation:** 3 good
**Contribution:** 3 good

**Summary:**

This paper proposes a novel idea by using a simple three-layer MLP, instead of fully connected layers, as the reflection stage to ask the network to reflect on its feed-forward decision by considering and evaluating all available choices/classes.

**Questions:**

1. How does the intuition behind getting Introspective Features different from traditional approaches with fully connected layers + softmax + cross-entropy loss? And why does the proposed approach works better?

2. Please clarify what does $\mathcal{H}(\cot)$ function stand for in equation 7.

3. What is the increased computational complexity of changing the last layer of the network into an Introspective Network?

**Strengths And Weaknesses:**

The proposed Introspection Network is very inspiring in transforming $N$ classes into $N$ possible introspective questions and answers. However, the robustness provided by the introspective networks (in Section 4) is not clearly presented.

---

> ### Author Response · Authors · 2022-08-03
> **Introspection intuition; Computational complexity**
>
> We thank the reviewer for their comments. We added to the complexity analysis in the paper and the changes are in line 778 in the revised paper.
>
> **Intuition for Introspection** Note that by feed-forward $f(\cdot)$ network in the paper, we refer to a regular network trained using cross-entropy (Lines 32-33). We name this network as a sensing network to differentiate it from the reflection stage. We justify our features both from a low-level mathematical and a high-level reasoning and cognitive science perspective.
>
> 1. *Mathematical intuition* We show that gradients of parameters are a function of only two logits for a well-trained network - the introspective class $y_I$ and the predicted class $y_{\hat{y}}$ (Lemma 1, Fig. 2). Hence, to mis-predict between $y_I$ and $y_{\hat{y}}$, noise has to alter the relationship between only these two weights so that the maximal projection in the output is from the mis-predicted weight. To remove this fragility, we extract $N$ such features. We train $\mathcal{H}(\cdot)$ and condition it on all these $N$ relationships. Hence, in our method, to create mis-predictions, the noise has to break $N$ such relationships. Hence, the same noise that disrupts the prediction of $f(\cdot)$ finds it harder to disrupt the prediction of $f(\cdot)$ and $\mathcal{H}(\cdot)$ together which is our two-stage inference. We summarize this in Lines 159-164. Moreover, we present ablation results in Table 9. We add additional MLP on the network itself without introspective features. In all cases, introspection provides better results.
>
> 2. *Cognitive science and reasoning perspective* By introspection, we are asking the network to rethink its decision based on contrastive questions - *Why P, rather than Q?*. This is a form of abductive reasoning that is used by humans to generalize to new situations. By new situations in a neural network, we refer to distributional differences in untrained scenarios. We expand on this in Appendix A, Lines (609-623) and (652-664). We consider untrained scenarios under the applications of robust recognition and calibration (Section 5 and 6), domain difference (C.6), Active Learning (Appendix D.1 and Table 2 in revised version), OOD (Appendix D.2 and Table 3 in revised paper), Image Quality Assessment (Appendix D.3), and uncertainty estimation (Appendix D.4).
>
> **Meaning of $\mathcal{H}(\cdot)$ in Eq.7** We thank the reviewer for this comment. While we had explicitly stated that $\mathcal{H}(\cdot)$ is an MLP function in the Introduction, we had not done so in Section 4. We added this in Line 183. The MLP in Fig.3 is $\mathcal{H}(\cdot)$.
>
> **Computational complexity** We discuss the computational complexity of the features themselves in the main paper. But due to space constraints the complexity of the MLP and its related ablation studies are presented in the Appendix.
>
> 1. *Structure of the MLP* The parameters and structure of the MLP is itself presented in Table 4. All the MLP networks used in various experiments of the paper are presented there. For a majority of the experiments in the main paper, we use ResNet-18 on CIFAR-10 and it replaces the $64\times 10$ final fully connected layer with an MLP that is $640\times 300 - 300\times 100 - 100\times 10$. Please note that this is an increase of around $2.23\times 10^5$ parameters. The initial feature size of 640 is because of concatenated introspective features from Theorem 1 (Line 177 onwards)
>
> 2. *Gains because of increase in parameter size* The above numbers are better put into context in Appendix C.2.1 and Fig. 8a. By increasing the parameter size by $2.23 \times 10^5$, we obtain the same results with an Introspective ResNet-18 as we would with a regular feed-forward ResNet-50. This is depicted in Fig. 8a where the performance gain (in red) of ResNet-18, matches that of the blue bar of ResNet-50. This is in addition to the calibration, OOD, uncertainty, and Active learning gains by the introspective network. Note that a ResNet-50 has more than double the parameters of a feed-forward ResNet-18, and the additional MLP is around 0.005% of the extra parameters required by a ResNet-50. We add this in blue in line 778. This analysis adds to the paper and we thank the reviewer for the comments.
>
> 3. *Ablation studies* A number of ablation studies are conducted in Table 9 and Appendix C.4.2. The effect of different structures of $\mathcal{H}(\cdot)$, adding the MLP of similar parameters directly on the logits, training jointly with a ResNet-18 and MLP are all presented and in all cases, the proposed introspective network with the introspective features show performance gains.

---

> > ### Comment · Reviewer_W6Yu · 2022-08-08
> > **Thanks for reply**
> >
> > Thanks for the detailed reply, I updated my score from 5 to 6.

---

### Official Review · Reviewer_aTYk · 2022-07-11

**Rating:** 7
**Confidence:** 3
**Soundness:** 4 excellent
**Presentation:** 4 excellent
**Contribution:** 4 excellent

**Summary:**

The authors propose a two-stage formulation for neural network decision making inspired by human introspection. The first stage is the regular discriminative feed-forward stage, where a deep network processes an input and generates the logit features. The introspective features for class I are then given as the change in the network parameters (up until the last layer) when a label of I is introduced for a sample x. The introspective features are then used to train a second introspective network, which tries to match the class predictions obtained from the introspective features to the ground truth predictions as well as the predictions from the first network. The authors provide theoretical derivations for these features, their approximation, sparsity, as well as how to extract them efficiently. The extensive experiments performed by the authors demonstrate that introspective learning improve neural network performance in terms of robustness, calibration, domain adaptation and active learning.

**Questions:**

1. In L43, the authors say "Consider any dataset with N classes. A network trained on this dataset will have N possible introspective questions and answers." Shouldn't such a network have N^2 possible introspective questions (Why class i, rather than j? where i, j ∈ {1, N})

2. L245 Typo: Robustness

**Limitations:**

The authors have adequately discussed the limitations of their work as well as its broader societal impact in Section 7.

**Strengths And Weaknesses:**

Strengths

1. While the idea of logit gradients encoding class similarity information is not new, using this information to improve robustness and calibration in neural networks is quite novel and very well motivated theoretically and empirically. The biological analogue drawn to Kahneman's Fast/Slow learning to motivate introspective learning is relevant, but somewhat tenuous.

2. The experiments conducted on robustness and calibration show that introspection helps improve both the out-of-distribution test accuracy for distorted images as well as the calibration of the neural network. The introspective neural networks tend to have very low expected calibration error in general.

3. The results in Table 1 show consistent improvement in performance on existing robustness techniques when using introspection on top. This motivates the usage of introspection as a general purpose technique to train more robust neural network models.

Weakness

1. This is not an issue with the work itself but against the nomenclature of the approach. The authors use a narrow definition of introspection pertinent specifically to discriminative learning : Why class X over class Y? In general, I am against the practice of co-adapting terms for ideas established in other fields (cognitive science in this case) and using them in an extremely narrow scope in machine learning. This proposed method covers one abductive reasoning based post-hoc 'introspection' approach, not all of introspection as it pertains to humans.

2. Could the low calibration error of the introspective model be affected by the fact that the introspective model is an MLP, whereas the models used for comparison are CNNs? [1] showed that vector/matrix scaling of CNN logits leads to better calibration error, and I wonder if the introspective model ends up doing something similar to reduce the calibration error.

References

[1] Chuan Guo, Geoff Pleiss, Yu Sun, and Kilian Q Weinberger, “On calibration of modern neural networks,” in International Conference on Machine Learning. PMLR, 2017, pp. 1321–1330

Disclaimer

I have not formally verified the proofs provided by the authors in Appendix

---

> ### Author Response · Authors · 2022-08-03
> **Usage of the term Introspection; Calibration as a function of limited network size**
>
> We thank the reviewer for their helpful comments and address some of the question below.
>
> **Narrow scope of proposed introspection** We agree with the reviewer that the method presented in the paper is itself quite narrow in scope when compared to human introspection. We mention this in line 69 where we define the framework of introspection as answers to only contrastive questions of the form *`Why P, rather than Q?'*. Even within such contrastive questions, we fix $P$ to be the prediction and contrast class $Q$ to be any one of $N$ classes. Hence there are $N$ introspective questions and answers. The reviewer alludes to possible $N\times N$ questions when $P$ itself can be altered. Prediction alteration requires interventions in data that we address in line 73 and in the Discussion section in line 379. In fact there can be even more introspective questions - $N\times 2^N$. We add an additional paragraph in the Appendix in line 685 under - A broader view on Introspection. We reproduce it below:
>
> *In this paper, we limit introspection as answers to* **Why P, rather than Q?** *questions. We limit $P$ to be the predictions made by networks. Hence, we are left with $N$ questions to answer. However, creative abduction calls for asking questions of the form* **Why P, rather than Q1 and Q2?**. *Such $Qs$ can extend to all $N$ classes. Hence, introspective labels can be the powerset of all one-hot labels - $2^N$. Moreover, the prediction itself can be made to change by intervening within data leading to questions of the form* **Why Q, rather than P?**. *These include counterfactual questions of the form* **What if?** *when considered from the output perspective. Hence, for $N$ classes, there can be $N\times 2^N$ introspective questions. Hence, the proposed features are only one possible feature set when considering introspection. However, we posit that all these features are a function of the data and the model, thereby making gradients an essential feature set while considering introspection and this paper provides intuitions as to their applicability.*
>
> With regards to nomenclature, we believe that the technique benefits from the usage of such terminologies as it fits the reasoning framework. It provides a means to explicitly examine the underlying model parameters to make a decision. It also creates interdisciplinary interest in the work and can lead to insightful discussions in the conference itself.
>
> **Calibration due to size and scaling of introspective MLP** As the reviewer notes, it is quite possible that the structured calibrated prediction technique from Eq. 8 in our method that is used to train $\mathcal{H}(\cdot)$ is similar to the matrix/vector scaling from [1]. Our analysis, inspired by [25 from the paper], assumes $\mathcal{H}(\cdot)$ to be the forecaster that is trained anew from the training set  (Line 190). The difference between the setting in [1] to ours is that we retrain $\mathcal{H}(\cdot)$ on the entire training set while the matrix/vector [1] scaling only uses the validation set to train a linear transformation on the predicted logits. We replicated the experiment from [1] by using $5000$ images of held-out data from CIFAR-10 trainset to train $\mathcal{H}(\cdot)$. However, accuracy results from introspection were worse because $\mathcal{H}(\cdot)$ was unable to train well with the smaller training set size. This is a weak empirical conclusion, since theoretically, both [1] and [25 from paper] perform the same forecasting with different size networks and the change is the training regimen itself.
>
> That being said, the introspective features themselves are robust. Applying a linear layer or an MLP on top of or in lieu of the last layer of the feed-forward network negatively affects generalizability. This is shown in the ablation studies of Table 9, Part 2. The results of the feed-forward network decreases to 63% from 67% when an additional MLP is added on top of the logits. Hence, while matrix/vector scaling helps calibration, it negatively affects the performance accuracy under distributional shift.
>
> We thank the reviewer for pointing out the typo and have addressed it in the revised version.

---

> > ### Comment · Reviewer_aTYk · 2022-08-04
> > **Response to author's rebuttal**
> >
> > Thanks a lot for taking the time to write a thoughtful rebuttal!
> >
> > 1. The explanation regarding possible counterfactuals for a N class classification problem and thus the increased cardinality of the possible set of introspective questions makes sense to me. My concern was less so about the exact possible problem size, and more so about defining the problem statement "**Why P, rather than Q?** where P is fixed to be the prediction and contrast class Q to be any one of N  classes" as introspection. However, I concede that this is an idealogical difference and does not take away from the overall quality of the paper.
> >
> > 2. This is a very interesting observation. I seem to have overlooked this conclusion regarding the generalizability and robustness of introspective features. So it seems like the introspective features are quite information dense in terms of class separability already, hence the fact that even going from a one layer MLP to a three layer MLP leads to worse generalizability. In hindsight, it makes sense since the sensing network is trained with a discriminative loss. The introspective features also seem 'better' in a way versus post-hoc calibration in that they not require a held-out validation set. For me, this reinforces the utility of obtaining introspective features from models and using them in the manner described by the authors.

---

### Official Review · Reviewer_Qogq · 2022-07-12

**Rating:** 4
**Confidence:** 3
**Soundness:** 2 fair
**Presentation:** 3 good
**Contribution:** 2 fair

**Summary:**

The paper introduces a novel 2-stage classification network where the 1st stage consists of a normal feed-forward pass through a network and the 2nd pass(Introspection) creates a set of features using gradients of the penultimate layer of feed-forward network and passes it through another MLP to predict the correct class. The set of features for each sample is created by concatenating the class-conditioned cross-entropy gradients at the penultimate layer for each class in the dataset. In other words, they calculate cross-entropy gradients for each sample by exhaustively setting every target class as the sample label. These gradients are then concatenated and passed through the MLP to generate the class label. They show that this approach gives them SOTA for Cifar10C and Cifar10 Cure datasets.

**Questions:**

SEE Strengths and Weaknesses section

**Limitations:**

Yes

**Strengths And Weaknesses:**

STRENGTHS:
1. The paper proposes a novel approach to rectify predictions for a classification network.
2. They show theoretically that the features calculated using gradients could be useful.
3. They show results on Cifar10C and Cifar10 Cure datasets.

WEAKNESSES:
1. This applicability of this work is currently limited to classification tasks that have very few target classes and a penultimate layer of modest dimension.
2. Results are shown only for Cifar10 related dataset and raises concerns if they work for other datasets like Flowers 102,Caltech 101, Cub200 etc.
3. Introspection stage is memory and compute-intensive. The authors should find ways to address this and other scaling issues

---

> ### Author Response · Authors · 2022-08-03
> **Additional applications - Active Learning, OOD, IQA, Uncertainty Estimation**
>
> We thank the reviewer for their constructive comments. We address each of the raised questions below.
>
> **Applicability of the work is limited to classification tasks that have very few target classes and a penultimate layer of modest dimension.** Because of limited space, we had to push the bulk of our experiments to the supplementary materials. Across Appendix C and D, we have multiple applications and ablation studies, including results on other datasets and networks.
>
> 1. In Appendix C.6, we show results for large-scale domain difference settings on Office and VisDA datasets. Office dataset has images of varying sizes including 1920x1080. Among both these datasets, introspection provides performance gains. Specifically on Office, we show results on ImageNet pre-trained ResNet-18, 34, 50, and 101 networks in Table 10. In Table 4, we added the MLP structure of $\mathcal{H}(\cdot)$ for different Office domains. The resulting dimensionality of the data when trained on Amazon domain is 7936x1 which is more than 12 times the size of ResNet-18 trained on CIFAR-10. The number of classes to discriminate between are 31. We show this to underscore our primary contribution that reflection is useful when there is a distributional shift between train and testsets, irrespective of the size of the images or the network used.
>
> 2. In Appendix D.1, we examine results in an active learning setting. All existing query strategies use the output logits to estimate the next batch of data to label. Hence, more calibrated and generalizable is the model, more effective is any query strategy. Akin to recognition experiments, all query strategies perform better under introspection on CIFAR-10C.
>
> 3. In Appendix D.2, we show results of Out-of-Distribution detection, when existing method like ODIN and baseline techniques are used on both feed-forward and introspective networks. We do so under two settings. The first is the normal OOD setting where the network is trained on CIFAR-10 and tested on one of four datasets – Textures, iSUN, Places365, and LSUN. The second setting is the harder adversarial setting where not only is the query data OD or ID, it is also under adversarial attack. In both these settings we see introspection providing performance gains.
>
> 4. In Appendix D.3, we examine human introspection on Image Quality Assessment (IQA). IQA requires humans to evaluate a noisy image based on its pristine version. Hence IQA most resembles our reflection setting. Datasets consist of noisy and pristine images, and quality scores. In IQA, we show that introspection provides statistically significant results on two datasets – MULTI-LIVE and TID2013.
>
> 5. In Appendix D.4, we show results on uncertainty estimation on rotated CIFAR-10R images. The best results are bayesian methods that use multiple models and outputs to create uncertainty measures. However, introspection (which is deterministic) provides comparable results to bayesian methods while performing better than TENT, another deterministic model.
>
> Since NeurIPS allows an additional page for the accepted version, we have added both the Active Learning and OOD experiments to the main paper. Please note that these are not new experiments but are taken from the supplementary materials that was submitted earlier.
>
> **Evaluation strategy across applications and using CIFAR-10** We believe that our framework is better evaluated through the setup where we use the same dataset for training across related applications. An important contribution of the paper is that we induce networks to be more robust during deployment. We look at this task of deployment holistically by considering multiple applications – recognition, calibration, OOD, Active Learning, and Uncertainty. We are unaware of any existing work that looks at all these applications together. This helps us make claims like the following on Line 224 - The same framework that robustly recognizes images despite noise can also detect noise to make an out-of-distribution detection. In literature, CIFAR-10 is the common dataset among all these applications. While CIFAR-10C and CIFAR-10-CURE are useful for recognition, CIFAR-10R is used for uncertainty estimation. Hence, we analyze any trained network holistically as opposed to methods that contribute to one application. For instance, AugMix technique produces more robust accuracy on CIFAR-10C. However, it does not help with calibration. Introspection on top of AugMix not only conserves accuracy, but also reduces calibration error by 43% (Line 315). Hence, a holistic look at distributional shift requires not just large-scale datasets, but also all facets of distributional deployment issues.
>
> This is not to say that larger datasets other than CIFAR-10 are immaterial. We also have large-scale images in Appendix C.6 and D.3 sections that show introspection on larger models and datasets. Appendix D.3 provides a completely new application that we believe most closely resembles introspection.

---

> > ### Author Response · Authors · 2022-08-03
> > **Memory and compute-intensive**
> >
> > From Lemma 1, and Fig. 2, we show that the majority of features in a well-trained network are sparse. This allows Theorem 1 which enables extracting introspective features in $\mathcal{O}(1)$ compute or time complexity. Since we only extract from the last layer, the feature extraction time is one forward pass through the network and one backward pass through the last layer (Lines 174-176).
> >
> > The memory or space complexity, however, is $\mathcal{O}(N \times d_{L-1})$. This is dependent on $N$. In Line 365, we explicitly acknowledge this under limitations and describe future work as one that removes the dependency on $N$. Currently, we have experimented with additional techniques that only backpropagate the top-K predictions. While the results on calibration and OOD are comparable, the results on recognition suffers. This is because of non-conformity of features and MLP's inability to be space-invariant - it requires the same features to be present across samples. For noisy images, the top-K predictions may be different.
> >
> > However, the algorithmic limitations are in line with reasoning systems that reason in both a fast and slow manner (Lines 371-375). We explicitly mention the use case of introspective networks to be under distributional shift (Line 201). This requires additional compute resources that may never be independent of $N$. The motivation from a biological perspective is provided in Lines 652 in the Appendix. We believe that this work adds to the literature and will create engaging discussions during the conference.

---

### Author Response · Authors · 2022-08-02
**Summary of revisions**

We thank the Reviewers for their feedback and comments. The revised paper is uploaded and the the changes are in blue. A summary of the major revisions are as follows:

1. We have added two applications where introspection is used as a plug-in approach from the supplementary to the main paper.

2. We have addressed all comments and questions raised by the reviewers.

The paper has been strengthened with these additions. We also address each of the raised questions and comments as a reply to each reviewer separately. We will be happy to address any additional concerns forthwith.

---

### Meta-Review · Area_Chair_C16R · 2022-08-29

**Recommendation:** Accept
**Confidence:** Certain

**Metareview:**

The paper proposes an approach for reflecting on model predictions in a classification task. The approach is novel and empirical evaluations show significant improvement over standard predictive networks. One of the reviewers is critical of the paper because this is not the first two-stage approach proposed. I do not think that this is fair criticism given the technical details of the current paper and an existing approach the reviewer points out is significantly different, as the reviewer themselves agree. Questions about the practicality of applicability when the number of classes are large and generalization of the conclusions to other datasets were raised, which are fair points in my opinion and addressing these points would make a stronger contribution.


**Award:**

No

---

### Decision · Program_Chairs · 2022-09-14

Accept